# Relational Verification Leaps Forward with RABBit

**Tarun Suresh**[1*] **Debangshu Banerjee**[1*] **Gagandeep Singh**[1,2]
[1]University of Illinois Urbana-Champaign, [2]VMware Research
{tsuresh3, db21, ggnds}@illinois.edu

## Abstract

We propose RABBit, a Branch-and-Bound-based verifier for verifying relational properties defined over Deep Neural Networks, such as robustness against universal adversarial perturbations (UAP). Existing SOTA complete $L_\infty$-robustness verifiers can not reason about dependencies between multiple executions and, as a result, are imprecise for relational verification. In contrast, existing SOTA relational verifiers only apply a single bounding step and do not utilize any branching strategies to refine the obtained bounds, thus producing imprecise results. We develop the first scalable Branch-and-Bound-based relational verifier, RABBit, which efficiently combines branching over multiple executions with cross-executional bound refinement to utilize relational constraints, gaining substantial precision over SOTA baselines on a wide range of datasets and networks. Code is at this URL.

## 1 Introduction

Deep neural networks (DNNs) are now widely used in safety-critical fields like autonomous driving and medical diagnosis [Amato et al., 2013], where their decisions can have serious consequences. However, understanding and ensuring their reliability is difficult due to their complex and opaque nature. Despite efforts to find and address vulnerabilities, such as adversarial attacks [Goodfellow et al., 2014, Madry et al., 2018, Moosavi-Dezfooli et al., 2017] and adversarial training techniques [Madry et al., 2018], ensuring safety remains a challenge. As a result, extensive research is focused on formally verifying the safety of DNNs. However, most of the existing $L_\infty$ robustness verification techniques can not handle relational properties common in practical situations. While significant efforts have been invested in verifying the absence of input-specific adversarial examples within the local neighborhood of test inputs, recent studies [Li et al., 2019a] emphasize that input-specific attacks are impractical regardless. Conversely, practical attack scenarios [Liu et al., 2023, Li et al., 2019b,a] involve the creation of universal adversarial perturbations (UAPs) [Moosavi-Dezfooli et al., 2017], which are crafted to impact a substantial portion of inputs from the training distribution. RaVeN [Banerjee et al., 2024b] and subsequently RACoon [Banerjee and Singh, 2024] showed that since the same adversarial perturbation is applied to multiple inputs, the executions on different perturbed inputs are related, exploiting the relationship between different executions significantly improves the precision of the verifier. Despite RaVeN and RACoon's ability to leverage cross-executional dependencies, both of them remain imprecise as they only apply a single bounding step and lack refinement using branching strategies used in SOTA complete non-relational verifiers.

**Key challenges:** For precise relational verification, we need efficient algorithms that can effectively combine branching strategies over multiple executions with bounding techniques that can leverage cross-executional dependencies. Theoretically, MILP (Mixed Integer Linear Programming) can exactly encode DNN executions with piecewise linear activation functions like ReLU over any input regions specified by linear inequalities. However, the associated MILP optimization problem is computationally expensive. For instance, encoding $k$ executions of a DNN with $n_r$ ReLU activations

---

*Equal contribution. Primary Correspondence: tsuresh3@illinois.edu

introduces $O(n_r \times k)$ integer variables in the worst case. As the cost of MILP optimization grows exponentially with the number of integer variables, even SOTA off-the-shelf solvers like Gurobi [Gurobi Optimization, LLC, 2018] struggle to verify small DNNs for a relational property over $k$ executions within a reasonable time limit. For scalability, SOTA non-relational verifiers like $\alpha, \beta$-CROWN [Wang et al., 2021] design custom "Branch and Bound" (BaB) solvers using more scalable differentiable optimization techniques such as gradient descent. However, these verifiers ignore dependencies between multiple executions, resulting in imprecise relational verification. Conversely, the SOTA relational verifier RACoon uses parametric linear relaxation for each activation to avoid integer variables and employs gradient descent to learn parameters that leverage cross-executional dependencies for verification. This method, however, introduces imprecision due to the replacement of non-linear activations with parametric linear approximations. Therefore, precise relational verification requires scalable algorithms that can: a) scale to the large DNNs used in this paper, b) effectively reduce imprecision from parametric linear relaxations, and c) utilize cross-executional dependencies.

**Our contributions:** We advance the state-of-the-art in relational DNN verification by:

- Efficiently combining branching strategies over multiple DNN executions with a cross-executional bounding method that utilizes dependencies between DNN's outputs from different executions while reducing imprecision resulting from parametric linear relaxations.
- Developing two "branch and bound" algorithms, each with its own advantages - **a) strong bounding:** applies cross-execution bounding at each step, branching over all executions. This method provides tighter bounds than RACoon (cross-executional bound refinement without branching) and $\alpha, \beta$-CROWN (branching without cross-executional bound refinement), **b) strong branching:** applies cross-execution bounding only at the start to derive fixed linear approximations for each execution. These approximations are then used to branch independently over each execution, exploring more branches per execution.
- Combining strong bounding and branching results into an efficiently optimizable MILP instance that leverages the benefits of both techniques, outperforming each individually.
- Performing extensive experiments on popular datasets and various DNNs (standard and robustly trained) to showcase the precision improvement over the current SOTA baselines.

## 2 Related Works

**Non-relational DNN verifiers:** Given a logical input specification $\phi$ and an output specification $\psi$, DNN verifiers formally prove that for all inputs $\mathbf{x}$ satisfying $\phi$, the output $N(\mathbf{x})$ of the DNN satisfies $\psi$. If the verification process fails, the verifier generates a counter-example where the output specification $\psi$ does not hold. DNN verifiers are broadly divided into three main types based on their ability to prove properties: - (i) sound but incomplete verifiers which may not always prove property even if it holds [Gehr et al., 2018, Singh et al., 2018, 2019b,a, Zhang et al., 2018, Xu et al., 2020, 2021], (ii) complete verifiers that can always prove the property if it holds [Wang et al., 2018, Gehr et al., 2018, Bunel et al., 2020a,c, Bak et al., 2020, Ehlers, 2017, Ferrari et al., 2022, Fromherz et al., 2021, Wang et al., 2021, Palma et al., 2021, Anderson et al., 2020, Zhang et al., 2022a] and (iii) verifiers with probabilistic guarantees [Cohen et al., 2019, Li et al., 2022]. Beyond the commonly studied $L_\infty$ robustness verification problem, several works adapt DNN verification techniques for specific applications, such as robustness against image rotation [Singh et al., 2019b, Balunovic et al., 2019], incremental verification [Ugare et al., 2023, 2024], interpretability [Banerjee et al., 2024a], and certifiable training [Mueller et al., 2023, Palma et al., 2024, Jiang and Singh, 2024].

**Relational DNN verifier:** Existing relational verifiers fall into two main categories based on the type of relational properties they can handle: (i) verifiers for properties such as UAP and fairness, which are defined across multiple executions of the same DNN [Zeng et al., 2023, Khedr and Shoukry, 2023, Meyer et al., 2024, Banerjee et al., 2024b, Banerjee and Singh, 2024, Banerjee et al., 2024c], and (ii) verifiers for properties like local DNN equivalence, defined over multiple executions of different DNNs on the same input [Paulsen et al., 2020, 2021]. For relational properties defined across multiple executions of the same DNN, existing verifiers [Khedr and Shoukry, 2023] reduce the verification problem to an $L_\infty$ robustness problem by constructing a "product DNN" that includes multiple copies of the same DNN. However, the relational verifier in [Khedr and Shoukry, 2023] treats all $k$ executions of the DNN as independent, which results in a loss of precision. On the other hand, [Zeng et al., 2023] (referred to as the I/O formulation) tracks the relationships between inputs used in multiple executions at the input layer, but it does not maintain the relationships between the

outputs fed into the subsequent hidden layers. As a result, it achieves only limited improvement over baseline verifiers that treat all executions independently. RaVeN [Banerjee et al., 2024b] uses DiffPoly a abstract intepretation based framework to track linear relationships between the outputs at all layers resulting from multiple executions of the same DNN. While RaVeN is significantly more precise than the I/O formulation, tracking linear constraints at each layer across all DNN executions can be computationally expensive. The SOTA relational verifier RACoon [Banerjee and Singh, 2024] improves the scalability of RaVeN while maintaining RaVeN's precision by introducing a new gradient-descent based bounding strategy called cross-executional bound refinement, as detailed in Section 3. There exist, probabilistic verifiers, [Xie et al., 2021, Zhang et al., 2022b] based on randomized smoothing [Cohen et al., 2019] for verifying relational properties. However, these works can only give probabilistic guarantees on smoothed models which have high inference costs. Similar to [Banerjee et al., 2024b, Banerjee and Singh, 2024], in this work, we focus on deterministic relational verifiers for DNNs with ReLU activation. However, RABBit can be extended to activations like Sigmoid with branching methods [Shi et al., 2024] and parametric bounds [Wu et al., 2023].

## 3 Preliminaries

We provide background on "branch and bound" (BaB) based non-relational DNN verification, as well as DNN safety properties that can be encoded as relational properties.

**Non-relational DNN verification:** For a single execution, non-relational DNN verification focuses on proving that, for all perturbations $\mathbf{x} + \boldsymbol{\delta}$ of a given input $\mathbf{x}$ specified by $\phi$, the network's output $\mathbf{y} = N(\mathbf{x} + \boldsymbol{\delta})$ meets a specified logical condition $\psi$. Commonly, safety properties such as $L_\infty$ robustness encode the output condition ($\psi$) as a linear inequality or a conjunction of linear inequalities over the DNN output $\mathbf{y} \in \mathbb{R}^{n_l}$. For instance, an output property could be expressed as $\psi(\mathbf{y}) = (\mathbf{c}^T \mathbf{y} \geq 0)$, where $\mathbf{c} \in \mathbb{R}^{n_l}$. Generally, even for DNNs with piecewise-linear activation functions and input constraints defined by linear inequalities, complete verification—i.e., always proving the property or finding a counterexample—is an NP-complete problem. Given a DNN $N : \mathbb{R}^{n_0} \to \mathbb{R}^{n_l}$ and a property defined by $(\phi, \psi)$, scalable yet sound (but incomplete) verifiers approximate the network's behavior by computing a linear approximation specified by $\mathbf{L} \in \mathbb{R}^{n_0}$ and $b \in \mathbb{R}$. For any input $\mathbf{x}$ satisfying $\phi$, this linear approximation ensures that $\mathbf{L}^T \mathbf{x} + b \leq \mathbf{c}^T N(\mathbf{x})$. The verifier then aims to show that $\mathbf{L}^T \mathbf{x} + b \geq 0$ for all $\mathbf{x}$ that satisfy $\phi$, which implies $\mathbf{c}^T N(\mathbf{x}) \geq 0$. While $\mathbf{L}^T \mathbf{x} + b$ provides a valid lower bound for $\mathbf{c}^T N(\mathbf{x})$, it may lack precision. To enhance this precision for piecewise-linear activations, state-of-the-art non-relational verifiers use a branch-and-bound (BaB) method. In each branching step, the problem is divided into smaller subproblems, while the bounding method computes a valid lower bound for each subproblem.

**Branching for piecewise linear activation:** The non-relational verifier computes $\mathbf{L}$ by replacing non-linear activations with linear relaxations, which introduces imprecision. However, for piecewise linear activations like ReLU, it is possible to consider each linear piece separately as different subproblems, avoiding the need for imprecise linear relaxations. For instance, for $y = ReLU(x)$, branching on $x$ and considering the cases $x \leq 0$ and $x \geq 0$ allows decomposing $ReLU(x)$ into two distinct linear pieces. Still in the worst case decomposing all ReLU nodes in a DNN results in exponential blowup making it practically infeasible. Therefore, SOTA non-relation verifiers like $\alpha, \beta$-CROWN [Wang et al., 2021] greedily pick a small subset of ReLU nodes for branching while using linear relaxations for the rest. We explain the bounding step used for each subproblem below.

**Bounding with parameter refinement:** Obtaining sound linear relaxations of activations $\sigma$ like ReLU, which are not used for branching, involves computing linear lower bounds $\sigma_l(x)$ and upper bounds $\sigma_u(x)$ that contain all possible outputs of $\sigma$ w.r.t all inputs $\mathbf{x}$ satisfying $\phi$. That is, for all possible input values $x$ of $\sigma$, $\sigma_l(x) \leq \sigma(x) \leq \sigma_u(x)$ holds. SOTA non-relational verifiers, such as $\alpha, \beta$-CROWN, improve precision by using parametric linear relaxations instead of static linear bounds and refine the parameters to facilitate verification of the property $(\phi, \psi)$. For example, for $ReLU(x)$, the parametric lower bound is $ReLU(x) \geq \alpha \times x$ with $\alpha \in [0, 1]$. Since $\alpha \times x$ remains a valid lower for any $\alpha \in [0, 1]$, this allows optimizing $\alpha$ while ensuring the bound remains mathematically correct. Each branched ReLU say $y = ReLU(x)$, introduces two subproblems each with one additional constraint $x \leq 0$ (or, $x \geq 0$) where ReLU behaves as a linear function i.e. $y = 0$ (or, $y = x$) respectively. To obtain the lower bound of $\mathbf{L}^T \mathbf{x} + b$ over inputs satisfying $\phi$ with the additional branching constraints $\alpha, \beta$-CROWN convert the constrained optimization problem into an unconstrained one by looking at the Lagrangian dual. The dual replaces each branching constraint by augmenting the minimization objective $\mathbf{L}^T \mathbf{x} + b$ with additional terms i.e. $\mathbf{L}^T \mathbf{x} + b + \beta^+ x$ for $x \leq 0$

or $\mathbf{L}^T\mathbf{x} + b + \beta^- x$ for $x \geq 0$ where $\beta^+ \geq 0$ and $\beta^- \leq 0$. Overall, at high level, $\alpha,\beta$-CROWN computes parametric linear approximations $\mathbf{L}(\boldsymbol{\alpha},\boldsymbol{\beta})^T\mathbf{x} + b(\boldsymbol{\alpha},\boldsymbol{\beta})$ and refine the parameters $\alpha,\beta$ to facilitate verification of $(\phi, \psi)$.

**DNN relational properties:** Relational properties defined for a DNN $N : \mathbb{R}^{n_0} \to \mathbb{R}^{n_l}$ defined over $k$ executions of $N$ are specified by the tuple $(\Phi, \Psi)$. Here, $\Phi : \mathbb{R}^{n_0 \times k} \to \{true, false\}$ (the input specification) encodes the input region $\Phi_t \subseteq \mathbb{R}^{n_0 \times k}$ encompassing all potential inputs corresponding to each of the $k$ executions of $N$. Furthermore, the safety property we expect the outputs of all $k$ executions of $N$ to satisfy is specified by $\Psi : \mathbb{R}^{n_l \times k} \to \{true, false\}$ (the output specification). Given $N$, an input specification $\Phi$ and an output specification $\Psi$, DNN relational verification seeks to formally prove whether $\forall \mathbf{x_1^*}, \ldots, \mathbf{x_k^*} \in \mathbb{R}^{n_0}.\Phi(\mathbf{x_1^*}, \ldots, \mathbf{x_k^*}) \implies \Psi(N(\mathbf{x_1^*}), \ldots N(\mathbf{x_k^*}))$ or otherwise provide a counterexample. The inputs to the $k$ executions of $N$ are denoted by $\mathbf{x_1^*}, \ldots, \mathbf{x_k^*}$ and the corresponding outputs are denoted by $N(\mathbf{x_1^*}), \ldots, N(\mathbf{x_k^*})$. For the $i$-th execution, commonly, the input region $\phi_t^i$ is a $L_\infty$ region around a fixed point $\mathbf{x_i} \in \mathbb{R}^{n_0}$ defined as $\phi_t^i = \{\mathbf{x_i^*} \in \mathbb{R}^{n_0} \mid \|\mathbf{x_i^*} - \mathbf{x_i}\|_\infty \leq \epsilon\}$ and the corresponding output specification $\psi^i(N(\mathbf{x_i^*})) = \bigwedge_{j=1}^m (\mathbf{c_{i,j}}^T N(\mathbf{x_i^*}) \geq 0)$. Consequently, $\Phi(\mathbf{x_1^*}, \ldots, \mathbf{x_k^*}) = \bigwedge_{i=1}^k (\mathbf{x_i^*} \in \phi_t^i) \bigwedge \Phi^\delta(\mathbf{x_1^*}, \ldots, \mathbf{x_k^*})$ where $\Phi^\delta(\mathbf{x_1^*}, \ldots, \mathbf{x_k^*})$ encodes the relationship between the inputs used in different execution and $\Psi(N(\mathbf{x_1^*}), \ldots, N(\mathbf{x_k^*})) = \bigwedge_{i=1}^k \psi^i(N(\mathbf{x_i^*}))$. Following this, we describe relational properties encoding important DNN safety configurations.

**UAP verification:** In a UAP attack, given a DNN $N$, the adversary aims to find an adversarial perturbation with a bounded $L_\infty$ norm that maximizes the rate at which $N$ misclassifies when the same adversarial perturbation is applied to all inputs from the distribution. The UAP verification problem aims to find the worst-case accuracy of $N$ against the UAP adversary. We refer to this worst-case accuracy as UAP accuracy in the rest of the paper. As shown by Theorem 2 in [Zeng et al., 2023], it is possible to statistically estimate the UAP accuracy of $N$ with respect to the input distribution if one can determine the UAP accuracy of $N$ on $k$ randomly selected images. We focus on the $k$-UAP verification problem for the rest of the paper as improving the precision of $k$-UAP verification directly improves the UAP accuracy on the input distribution [Banerjee and Singh, 2024]. The $k$-UAP verification problem is fundamentally different from local $L_\infty$ robustness verification since the same adversarial perturbation is applied across the set of inputs. Thus, improving precision for the UAP verification problem requires a relational verifier that can exploit dependencies between the perturbed inputs. We provide the $\Phi$ and $\Psi$ of the UAP verification problem in Appendix A.1.

## 4 Cross-executional BaB

The key distinction between relational and non-relational DNN verification is the dependency between different DNN executions, which necessitates that any precise relational verifier utilizes these cross-execution dependencies. For instance, for $k$-UAP problem with two images $\mathbf{x_1}$, $\mathbf{x_2}$ consider the scenario where both $\mathbf{x_1}$ and $\mathbf{x_2}$ have valid adversarial perturbations $\boldsymbol{\delta_1}$ and $\boldsymbol{\delta_2}$ but no common perturbation say $\boldsymbol{\delta}$ that works for both $\mathbf{x_1}$ and $\mathbf{x_2}$. In this case, any non-relational verification that does not account for cross-execution dependencies can never prove the absence of a common perturbation given that both $\mathbf{x_1}$, $\mathbf{x_2}$ have valid adversarial perturbations. This highlights the importance of utilizing cross-executional dependencies. The SOTA relational verifier RACoon [Banerjee and Singh, 2024] leverages cross-execution dependencies to **jointly** optimize the $\boldsymbol{\alpha}$ parameters from different executions, significantly improving the precision of relational verification. However, RACoon only uses parametric linear relaxations for non-linear activations and lacks a branching step, resulting in reduced precision, as confirmed by our experimental results in Section 6. To address this, we propose two separate BaB algorithms, each with its benefits, described in Sections 4.1 and 4.2. Finally, we combine the results to formulate an efficiently optimizable MILP instance in Section 5

### 4.1 Strong Bounding

Before going into the details, we briefly review the cross-executional bound refinement proposed in RACoon. For $k$-UAP, given any subset $S$ of the $k$ executions, RACoon can verify the absence of any common perturbation that works for **all** executions in $S$ with cross-executional bound refinement. For all $i \in S$, let $(\mathbf{L}_i(\boldsymbol{\alpha}_i), \mathbf{b}_i(\boldsymbol{\alpha}_i))$ denote the parametric linear approximations corresponding to the $i$-th execution. Then the optimal value $t^* = \max_{\boldsymbol{\alpha}_i, \lambda_i} -\epsilon \times \| \sum_{i \in S} \lambda_i \times \mathbf{L}_i(\boldsymbol{\alpha}_i)\|_1 + \sum_{i \in S} \lambda_i \times a_i(\boldsymbol{\alpha}_i) \geq 0$ proves the absence of a common perturbation $\boldsymbol{\delta}$ for $S$. Here, $\epsilon$ is the perturbation bound i.e. $\|\boldsymbol{\delta}\|_\infty \leq \epsilon$, $a_i(\boldsymbol{\alpha}_i) = \mathbf{b}_i(\boldsymbol{\alpha}_i) + \mathbf{L}_i(\boldsymbol{\alpha}_i)^T\mathbf{x_i}$ and $\lambda_i \in [0, 1]$ with $\sum_{i \in S} \lambda_i = 1$ are the cross-executional parameters that relate linear approximations from different execution enabling joint optimization over $\boldsymbol{\alpha}_i$s. Next,

we detail the first BaB method - strong bounding that combines cross-executional bounding with branching methods to verify the absence of common perturbation for any subset of $n = |S|$ executions.

**Branching and bounding:** For $n \leq k$ executions, we construct a "product DNN" by duplicating the DNN $n$ times, one for each execution. Formally, a product DNN is a function $N^n : \mathbb{R}^{n_0 \times n} \to \mathbb{R}^{n_l \times n}$ with $N^n(\mathbf{x_1}, \ldots, \mathbf{x_n}) = [N(\mathbf{x_1}), \ldots, N(\mathbf{x_n})]^T$. At each branching step, we greedily select a subset of unbranched ReLU activations from the product DNN and branch on them, while using parametric linear relaxations for the rest. We adapt existing greedy branching heuristics, such as BaBSR [Bunel et al., 2020b], for selecting the candidate ReLU activations. The heuristic computes a score for each unbranched ReLU activation in the product DNN, and we branch on the activations with the highest scores. Next, we detail the bounding method applied to each subproblem resulting from branching. Since the number of subproblems can be large, the bounding method needs to be fast yet capable of leveraging both branching constraints and cross-executional dependencies. However, the cross-executional bound refinement from RACoon can not handle branching constraints, while the bounding step from $\alpha, \beta$-CROWN does not utilize dependencies across executions. Hence, we develop a three-step algorithm for obtaining the optimal value $t^*$ with fast gradient descent-based methods. First, we replace these branching constraints by introducing dual variables $\boldsymbol{\beta}$, resulting in new parametric linear approximations $(\mathbf{L}_i(\boldsymbol{\alpha}_i, \boldsymbol{\beta}_i), b_i(\boldsymbol{\alpha}_i, \boldsymbol{\beta}_i))$ for each subproblem for all $i \in S$. Then for each subproblem, we introduce additional variables $\lambda_i$ for each execution with constraints $\lambda_i \in [0, 1]$ and $\sum_{i \in S} \lambda_i = 1$. These $\lambda_i$s relate linear approximations from different executions capturing cross-executional dependencies. This reduces finding $t^*$ for each subproblem to the following optimization problem $t^* = \max_{\boldsymbol{\alpha}_i, \boldsymbol{\beta}_i, \lambda_i} -\epsilon \times \| \sum_{i \in S} \lambda_i \times \mathbf{L}_i(\boldsymbol{\alpha}_i, \boldsymbol{\beta}_i)\|_1 + \sum_{i \in S} \lambda_i \times a_i(\boldsymbol{\alpha}_i, \boldsymbol{\beta}_i)$. Here, $a_i(\boldsymbol{\alpha}_i, \boldsymbol{\beta}_i) = b_i(\boldsymbol{\alpha}_i, \boldsymbol{\beta}_i) + \mathbf{L}_i(\boldsymbol{\alpha}_i, \boldsymbol{\beta}_i)^T \mathbf{x_i}$. Finally, we apply projected gradient ascent to refine parameters $(\boldsymbol{\alpha}_i, \boldsymbol{\beta}_i, \lambda_i)$. The detailed derivation of the bounding step and the proof of correctness is in Appendix B. Precision gains of strong bounding over the baselines are in Section 6.2. Suppose, $\mathcal{F}(S)$ denotes the set of subproblems then Theorem 4.1 proves the absence of common perturbation for the subset of executions $S$.

**Theorem 4.1.** *If* $\min_{\mathcal{F}(S)} \max_{\boldsymbol{\alpha}_i, \boldsymbol{\beta}_i, \lambda_i} -\epsilon \times \| \sum_{i \in S} \lambda_i \times \mathbf{L}_i(\boldsymbol{\alpha}_i, \boldsymbol{\beta}_i)\|_1 + \sum_{i \in S} \lambda_i \times a_i(\boldsymbol{\alpha}_i, \boldsymbol{\beta}_i) \geq 0$ *then executions in $S$ do not have a common perturbation* $\boldsymbol{\delta} \in \mathbb{R}^{n_0}$ *with* $\|\boldsymbol{\delta}\|_\infty \leq \epsilon$.

**Proof:** The detailed proof is in the Appendix B.

While strong bounding effectively combines cross-executional refinement with branching, it has the following drawbacks that led to the development of the 2nd BaB method. First, strong bounding branches over all executions simultaneously, which limits the number of branches explored per execution within a fixed timeout compared to branching on individual executions. For instance, if strong bounding solves $m$ subproblems for $n$ executions, then assuming each execution branched uniformly, each execution gets only $m^{\frac{1}{n}}$ subproblems. In contrast, given the same timeout, branching individually allows exploring $\frac{m}{n}$ subproblems per execution. Second, strong bounding only proves the absence of common perturbation, a relaxation of the $k$-UAP problem. To mitigate this, RACoon uses parameter refinement to obtain linear approximations and formulate a MILP, providing a more precise bound on $k$-UAP accuracy. However, for strong bounding, as the number of subproblems increases and each subproblem has a different linear approximation, formulating a MILP with each linear approximation is practically infeasible. Restricting the number of linear approximations can help accommodate MILP formulation by compromising on the strong cross-executional bounding.

### 4.2 Strong Branching

Unlike strong bounding, strong branching explores more branches by branching on each execution independently. Additionally, for each execution, we aim to keep the number of linear approximations small post-branching, ensuring the MILP instance using these approximations remains easy to optimize. To limit the number of linear approximations for each execution $i$, we fix a set of linear coefficients $\{\mathbf{L}_1, \ldots, \mathbf{L}_m\}$ called "target coefficients" and for each $j \in [m]$, $\mathbf{L}_j \in \mathbb{R}^{n_0}$ compute valid lower bound $b_j^*$ of the following optimization problem $\min_{\boldsymbol{\delta}} \mathbf{c}^T N(\mathbf{x_i} + \boldsymbol{\delta}) - \mathbf{L}_j^T(\mathbf{x_i} + \boldsymbol{\delta})$ with $\|\boldsymbol{\delta}\|_\infty \leq \epsilon$ using BaB. In this case, for all $\boldsymbol{\delta} \in \mathbb{R}^{n_0}$ with $\|\boldsymbol{\delta}\|_\infty \leq \epsilon$ the refined bias $b_j^*$ and $\mathbf{L}_j$ remain a valid lower bound of $\mathbf{c}^T N(\mathbf{x_i} + \boldsymbol{\delta})$ i.e. $\mathbf{L}_j^T(\mathbf{x_i} + \boldsymbol{\delta}) + b_j^* \leq \mathbf{c}^T N(\mathbf{x_i} + \boldsymbol{\delta})$. Moreover, since we only refine the bias, the number of linear approximations remains the same as at the start of BaB, even after branching. Next, we describe how we utilize cross-execution dependencies while branching on each execution independently.

**Selecting targets:** To select target coefficients, we greedily pick subsets of executions and run cross-executional refinement from RACoon without branching on each subset of executions. We describe the greedy selection strategy in Section 5. For each set of executions, we add the linear approximations obtained by cross-executional refinement to the corresponding executions' target sets. Cross-executional refinement ensures for each execution set the parameters corresponding to the linear approximations are tailored for the relational verification.

**Bounding and branching:** Given a target coefficient $\mathbf{L}_t \in \mathbb{R}^{n_0}$, since finding the exact solution of $\min_{\boldsymbol{\delta}} \mathbf{c}^T N(\mathbf{x_i} + \boldsymbol{\delta}) - \mathbf{L}_t^T(\mathbf{x_i} + \boldsymbol{\delta})$ is computationally expensive, strong branching aims to obtain a tight mathematically correct lower bound on the difference $\mathbf{c}^T N(\mathbf{x_i} + \boldsymbol{\delta}) - \mathbf{L}_t^T(\mathbf{x_i} + \boldsymbol{\delta})$. For any subproblem, let $(\mathbf{L}(\boldsymbol{\alpha}, \boldsymbol{\beta}), b(\boldsymbol{\alpha}, \boldsymbol{\beta}))$ denote the parametric linear approximation. Then for this particular subproblem, for all $\boldsymbol{\alpha}, \boldsymbol{\beta}$, $\mathbf{L}(\boldsymbol{\alpha}, \boldsymbol{\beta})^T(x_i + \boldsymbol{\delta}) + b(\boldsymbol{\alpha}, \boldsymbol{\beta}) \leq \mathbf{c}^T N(\mathbf{x_i} + \boldsymbol{\delta})$ and subsequently:

$$\max_{\boldsymbol{\alpha}, \boldsymbol{\beta}} \min_{\|\boldsymbol{\delta}\|_\infty \leq \epsilon} (\mathbf{L}(\boldsymbol{\alpha}, \boldsymbol{\beta}) - \mathbf{L}_t)^T(\mathbf{x_i} + \boldsymbol{\delta}) + b(\boldsymbol{\alpha}, \boldsymbol{\beta}) \leq \min_{\|\boldsymbol{\delta}\|_\infty \leq \epsilon} \mathbf{c}^T N(\mathbf{x_i} + \boldsymbol{\delta}) - \mathbf{L}_t^T(\mathbf{x_i} + \boldsymbol{\delta}) \quad (1)$$

The optimal solution of the max-min problem in Eq. 1 provides a mathematically correct lower bound of $\min_{\boldsymbol{\delta}} \mathbf{c}^T N(\mathbf{x_i} + \boldsymbol{\delta}) - \mathbf{L}_t^T(\mathbf{x_i} + \boldsymbol{\delta})$ for each subproblem. However, it is hard to solve a max-min problem with scalable differentiable optimization techniques like gradient descent typically used for large DNNs considered in this paper. Instead, we compute a closed form of the inner minimization problem reducing the optimization instance to a more tractable maximization problem (Theorem 4.2).

**Theorem 4.2.** *For any $\boldsymbol{\alpha}, \boldsymbol{\beta}$, if $\mathbf{L}(\boldsymbol{\alpha}, \boldsymbol{\beta}) \in \mathbb{R}^{n_0}$ and $b(\boldsymbol{\alpha}, \boldsymbol{\beta}) \in \mathbb{R}$ then $\min_{\|\boldsymbol{\delta}\|_\infty \leq \epsilon}(\mathbf{L}(\boldsymbol{\alpha}, \boldsymbol{\beta}) - \mathbf{L}_t)^T(\mathbf{x} + \boldsymbol{\delta}) + b(\boldsymbol{\alpha}, \boldsymbol{\beta}) = -\epsilon \times \|\mathbf{L}(\boldsymbol{\alpha}, \boldsymbol{\beta}) - \mathbf{L}_t\|_1 + (\mathbf{L}(\boldsymbol{\alpha}, \boldsymbol{\beta}) - \mathbf{L}_t)^T\mathbf{x} + b(\boldsymbol{\alpha}, \boldsymbol{\beta}).$*

**Proof:** The proof is in Appendix C.

We apply a projected gradient ascent to optimize the maximization with the closed form obtained above (Appendix C.1). The proof of the correctness of the bounding method is in Appendix C. Note the proof of correctness does not necessitate the optimizer to find the global optimum. This is important since gradient ascent may not always converge to the global optimum. Since strong branching branch on each execution independently we reuse the branching strategy of $\alpha, \beta$-CROWN.

## 5 RABBit

In this section, we detail the algorithm (Algo. 1) that combines the results from strong bounding and strong branching to formulate the MILP. Running strong bounding on all $2^k - 1$ non-empty subsets of $k$ executions is impractical. Therefore, we use a greedy approach to select subsets of executions for strong bounding. Similarly, for strong branching, we greedily select the target linear coefficients. First, we describe both greedy strategies before moving on to the MILP formulation.

**Elimination of individually verified executions:** RABBit maintains a list of unverified indices and eliminates any executions that can be verified individually and does not consider them for subsequent steps (lines 3, 8, and 13 in Algo. 1). For instance, for $k$-UAP verification, we do not need to consider those executions that are proved to have no adversarial perturbation $\boldsymbol{\delta}$ such that $\|\boldsymbol{\delta}\|_\infty \leq \epsilon$. Pruning individually verified executions improves the runtime without any compromise on the precision of the relational verifier (see Theorem B.1 [Banerjee and Singh, 2024]).

**Greedy target coefficient selection:** RABBit first runs RACoon which in turn executes an incomplete non-relational verifier $\alpha$-CROWN [Xu et al., 2021] eliminating the verified executions (line 8 in Algo. 1). Subsequently, for target selection, RABBit greedily picks the first $k_t$ (hyperparameter) executions based on $s_i$ the lower bound on $\psi^i(N(\mathbf{x_i} + \boldsymbol{\delta}))$ as computed by $\alpha$-CROWN, prioritizing executions with higher $s_i$ (line 9). Intuitively, for unverified executions, $s_i$ measures the maximum violation of the output specification $\psi^i(N(\mathbf{x_i} + \boldsymbol{\delta}))$ and thus leads to the natural choice of picking executions with smaller violations. For each selected execution $i$, we choose up to $m$ target coefficients by iterating over all subsets $i \in S$ considered by RACoon, and selecting linear approximations corresponding to the top $m$ subsets. The cross-executional lower bound $t^*$ from RACoon decides the priority of each subset $S$. Subsets $S$ with higher $t^*$ indicate smaller violations and are more likely to be verified for the absence of a common perturbation, making them suitable for target selection.

**Selection of subsets of executions for strong bounding:** Thereafter, until timeout $\zeta$, we run strong bounding on subsets of executions from individually unverified executions $I$. For each subset $S \subseteq I$,

**Algorithm 1** RABBit

1: **Input:** $N$, $(\Phi, \Psi)$, $k$, $k_t$, timeout $\zeta$
2: **Output: M.**
3: $I \leftarrow \{\}$              $\triangleright$ Unverified indices
4: $\mathcal{L} \leftarrow \{\}$         $\triangleright$ Linear approximations
5: $C \leftarrow \{\}$         $\triangleright$ Cross-verified executions
6: $\mathbf{s} \leftarrow \{\}$      $\triangleright$ Lower bounds from $\alpha$-Crown
7: $\mathbf{M} \leftarrow 0$    $\triangleright$ Initialize verified UAP accuracy
8: $(I, \mathcal{L}, C, \mathbf{s}) \leftarrow \text{RACoon}(N, (\Phi, \Psi), k)$
9: $I_1 \leftarrow$ top-$k_t$ indices from $I$ based on $\mathbf{s}$
10: **for** $i \in I_1$ **do**
11:      $b_i^* \leftarrow \text{StrongBranching}(\phi^i, \psi^i, \mathcal{L}[i])$
12:      **if** $\text{Verified}(\phi^i, \psi^i, \mathcal{L}[i], b_i^*)$ **then**
13:          $I \leftarrow I \setminus \{i\}$
14:      **end if**
15:      $\text{UpdateBias}(\mathcal{L}[i], b_i^*)$

16:      $\mathcal{M} \leftarrow \text{MILP}(\mathcal{L}, \Phi, \Psi, k, I, C)$
17:      $\mathbf{M} \leftarrow \max{(\mathbf{M}(\Phi, \Psi), \text{Opt}(\mathcal{M}))}$
18: **end for**
19: $I_2 \leftarrow$ top-$k_t$ indices from $I$ based on $\mathbf{s}$
20: **while** $\text{time}() < \zeta$ **do**
21:      $S \leftarrow$ Greedily select subset of $I_2$
22:      $t_S \leftarrow \text{StrongBounding}(S, \Phi, \Psi)$
23:      **if** $t_S \geq 0$ **then**
24:          $C \leftarrow \text{Append}(C, S)$
25:          $\mathcal{M} \leftarrow \text{MILP}(\mathcal{L}, \Phi, \Psi, k, I, C)$
26:          $\mathbf{M} \leftarrow \max{(\mathbf{M}, \text{Opt}(\mathcal{M}))}$
27:      **end if**
28: **end while**
29: **return M**

the cross-executional bound obtained by RACoon on $S$ decides its priority. However, considering all non-empty subsets of $I$ can be expensive. Instead, similar to strong branching, we first pick top-$k_t$ executions ($I_2$) from $I$ (Algo 1 line 19). We sort all non-empty subsets $S \subseteq I_2$ based on their priority and, in each iteration, run strong bounding on the highest-priority subset that has not been scheduled yet (Algo 1 line 22). Given a large timeout, RABBit would eventually select all subsets from $I_2$.

**MILP Formulation:** The MILP formulation uses both the refined biases from strong branching (line 11) and the subsets $S$ of executions verified for the absence of common perturbation from strong bounding (line 22) to compute final verified UAP accuracy. RABBit MILP formulation involves three steps. First, we deduce linear constraints between the input and output of $N$ for each unverified execution using linear approximations of $N$ with refined bias obtained by strong branching. Secondly, we add constraints for each subset $S$ verified for the absence of common perturbation with strong bounding. Then, similar to the current SOTA baseline [Banerjee and Singh, 2024], we encode the output specification $\Psi$ as a MILP objective, introducing only $O(k)$ integer variables. Finally, we use an off-the-shelf MILP solver [Gurobi Optimization, LLC, 2018] to optimize the MILP.

$\Psi$ **encoding:** First, we show the MILP objective $\mathbf{M}$ that encodes $\Psi$. We introduce binary variables $z_i \in \{0, 1\}$ for each individually unverified execution in $I$ where for any perturbation $\boldsymbol{\delta} \in \mathbb{R}^{n_0}$ and $\|\boldsymbol{\delta}\|_\infty \leq \epsilon$, $z_i = 1$ implies $\psi^i(N(x_i + \boldsymbol{\delta})) = True$. Then the finding the worst case UAP accuracy is equivalent to the following $\mathbf{M} = \frac{1}{k} \times \left((k - |I|) + min_{\|\delta\|_\infty \leq \epsilon} \sum_{i \in I} z_i\right)$.

**Constraints encoding:** We add constraints from strong bounding, strong branching, and from the linear approximation obtained from the call to RACoon (Algo. 1 line 8). Suppose for any subset $S \subseteq I$, strong bounding verifies the absence of common perturbation. Then for all $\boldsymbol{\delta} \in \mathbb{R}^{n_0}$ and $\|\boldsymbol{\delta}\|_\infty \leq \epsilon$ at least one of the executions from $S$ will always satisfy the corresponding output specification. Hence, for every such $S$ we add the constraint: $\sum_{i \in S} z_i \geq 1$. Now, let for any $i \in I$, $\{(\mathbf{L}_i^1, b_i^1), \ldots, (\mathbf{L}_i^m, b_i^m)\}$ denote set of linear approximation with $b_i^m$ either coming from RACoon or from strong branching. Then we add the following constraints $z_i \geq z_i'$, $z_i' = (o_i \geq 0)$, $o_i \geq \mathbf{L}_i^{jT}(\mathbf{x_i} + \boldsymbol{\delta}) + b_i^j$ where $j \in [m]$, and $o_i \in \mathbb{R}$, $z_i'$ are new real and integer variables respectively.

**Limitations:** Although RABBit outperforms SOTA verifiers in relational verification, like all deterministic verifiers, whether relational or non-relational (including ours), do not scale to deep neural networks (DNNs) trained on very large datasets such as ImageNet. RABBit is sound but incomplete, meaning it may not be able to prove certain relational properties even if they are true. Note that all complete non-relational verifiers are also incomplete for relational properties since they do not track any dependencies between executions.

## 6 Experimental Evaluation

We evaluate the effectiveness of RABBit on multiple relational properties such UAP accuracy ( Table 1) and top-k accuracy (Appendix Table 5), DNNs, and datasets. In our evaluation, we compare

Table 1: RABBit Efficacy Analysis for Worst-Case UAP Accuracy

| Dataset | Network Structure | Training Method | Perturbation Bound ($\epsilon$) | CROWN | $\alpha-$CROWN | $\alpha,\beta-$CROWN | MN-BaB | GCP-CROWN | I/O | RACoon | Strong Bounding | Strong Branching | RABBit |
|---|---|---|---|---|---|---|---|---|---|---|---|---|---|
| CIFAR10 | ConvSmall | Standard | 1/255 | 44.8 | 45.4 | 62.2 | 55.0 | 61.8 | 45.4 | 45.4 | 63.8 (+1.6) | 63.2 (+1.0) | 65.4 (+3.2) |
| | ConvSmall | DiffAI | 5/255 | 44.4 | 49.6 | 53.8 | 55.0 | 53.8 | 50.4 | 51.6 | 59.0 (+4.0) | 59.0 (+4.0) | 59.8 (+4.8) |
| | ConvSmall | SABR | 2/255 | 75.2 | 75.8 | 79.4 | 80.0 | 80.0 | 76.8 | 78.2 | 83.0 (+3.0) | 83.8 (+3.8) | 84.0 (+4.0) |
| | ConvSmall | CITRUS | 2/255 | 74.8 | 76.0 | 79.2 | 79.6 | 79.6 | 77.0 | 78.8 | 82.8 (+3.2) | 83.2 (+3.6) | 83.6 (+4.0) |
| | ConvBig | DiffAI | 2/255 | 46.6 | 51.8 | 61.2 | 61.6 | 61.2 | 53.2 | 54.8 | 62.8 (+1.2) | 62.6 (+1.0) | 63.0 (+1.6) |
| MNIST | ConvSmall | Standard | 0.07 | 53.0 | 59.4 | 83.6 | 77.4 | 84.2 | 60.0 | 60.6 | 84.2 (+0.0) | 84.2 (+0.0) | 84.8 (+0.6) |
| | ConvSmall | DiffAI | 0.13 | 51.8 | 57.0 | 76.6 | 77.0 | 77.0 | 57.2 | 58.4 | 79.0 (+2.0) | 78.6 (+1.6) | 80.0 (+3.0) |
| | ConvSmall | SABR | 0.15 | 27.0 | 38.0 | 50.4 | 51.2 | 60.2 | 42.2 | 45.8 | 62.6 (+2.4) | 62.2 (+2.0) | 63.4 (+3.2) |
| | ConvSmall | CITRUS | 0.15 | 28.8 | 41.6 | 73.0 | 69.2 | 73.0 | 41.6 | 44.6 | 74.0 (+1.0) | 73.4 (+0.4) | 74.6 (+1.6) |

RABBit against SOTA baselines, including non-relational verifiers CROWN [Zhang et al., 2018], $\alpha$-CROWN [Xu et al., 2021], $\alpha, \beta$-CROWN [Wang et al., 2021], MN-BaB [Ferrari et al., 2022], GCP-CROWN [Zhang et al., 2022a], as well as relational verifiers I/O Formulation [Zeng et al., 2023] and RACoon. As previously noted, RaVeN adds linear constraints for each layer, which restricts its scalability as the number of executions $k$ increases. Therefore, we compare RABBit with RaVeN for a smaller execution count of $k = 5$, as shown in Appendix Table 4. Additionally, we show that: a) given the same time, RABBit always outperforms the SOTA BaB-based non-relational verifier $\alpha, \beta$-CROWN; b) strong bounding computes a tighter bound on $t^*$ than $\alpha, \beta$-CROWN; and c) we provide an ablation study on $\epsilon$ and $k$ used by RABBit.

## 6.1 Experiment Setup

**Networks**. We use standard convolutional architectures, such as ConvSmall and ConvBig, which are used to evaluate both SOTA relational [Banerjee and Singh, 2024] and non-relational verifiers [Wang et al., 2021] (see Table 1). We provide the details of the DNN architectures in the Appendix D.1. We use networks trained using both standard training methods and robust training strategies, such as DiffAI [Mirman et al., 2018], SABR [Mueller et al., 2023], and CITRUS [Xu and Singh, 2024]. Our experiments utilize publicly available pre-trained DNNs sourced from the CROWN repository [Zhang et al., 2020], $\alpha, \beta$-CROWN repository [Wang et al., 2021], and ERAN repository [Singh et al., 2019b]. The clean accuracies of these networks are reported in Appendix D.2.

**Implementation details and hyperparameters**. We implemented our method in Python with Pytorch V1.11 on top of SOTA complete non-relational verifier $\alpha, \beta$-CROWN [Wang et al., 2021]. We used Gurobi V11.0 as the off-the-shelf MILP solver. For both strong bounding and strong branching, we use Adam [Kingma and Ba, 2014] for parameter learning and run it for 20 iterations on each subproblem. We set the value of $k_t = 10$ for CIFAR-10 and $k_t = 24$ for MNIST networks respectively. We use a single NVIDIA A100-PCI GPU with 40 GB RAM for bound refinement and an Intel(R) Xeon(R) Silver 4214R CPU @ 2.40GHz with 64 GB RAM for MILP optimization. For any relational property with $k$ executions, we give an overall timeout of $k$ minutes (averaging 1 minute/execution) to RABBit and all baselines. Each MILP instance gets a timeout of 10 minutes. We issue the MILP optimization call on line 25 of Algo. 1 in a separate thread for runtime optimization, ensuring that the MILP optimization process does not unnecessarily block the subsequent iterations of the while loop (line 20 of Algo. 1).

## 6.2 Experimental Results

**Effectiveness of RABBit:** Table 1 compares the results of RABBit to all baselines across different datasets (column 1) and DNN architectures (column 2) trained with various methods (column 3), with $\epsilon$ values defining the $L_\infty$ bound of $\boldsymbol{\delta}$ in column 4. For each DNN and $\epsilon$, we run RABBit and all the baselines on 10 relational properties each defined with $k = 50$ randomly selected inputs, and report the worst-case UAP accuracy averaged over the 10 properties. Note that for each DNN, we exclude inputs misclassified by the DNN. We compare the performance of RABBit against SOTA relational and complete non-relational verifiers as well as against strong bounding and strong branching.

The results in Table 1 demonstrate that strong bounding, strong branching, and RABBit all outperform the existing SOTA verifiers on all DNNs and $\epsilon$. Notably, RABBit gains up to +4.8% and up to +3.2% improvement in the worst-case UAP accuracy (averaged over 10 runs) for CIFAR10 and MNIST DNNs, respectively. RABBit also efficiently scales to the largest verifiable DNN architectures such as ConvBig, conferring up to +1.6% improvement in worst-case UAP accuracy. In some cases, strong bounding outperforms strong branching, while in others, strong branching outperforms strong

bounding, highlighting the importance of both methods. RABBit combines the strengths of both strong branching and strong bounding, producing the best results overall.

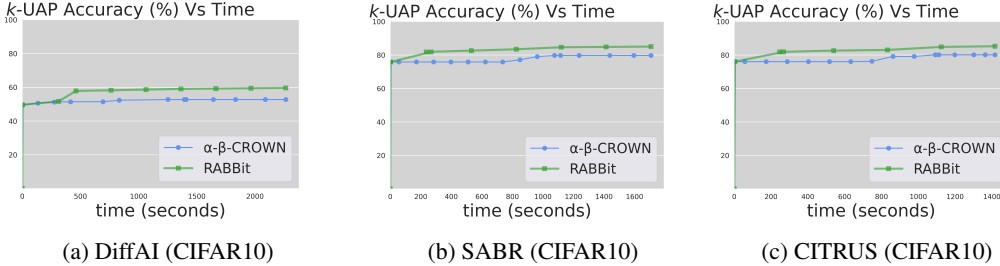

(a) DiffAI (CIFAR10)  (b) SABR (CIFAR10)  (c) CITRUS (CIFAR10)

Figure 1: Average Worst Case $k$-UAP accuracy vs Time for ConvSmall CIFAR10 DNNs.

**Time vs UAP Accuracy Analysis:** Fig. 1 shows timewise the worst-case UAP accuracy (averaged over 10 runs) for different ConvSmall CIFAR10 networks with $k = 50$ on $\epsilon$ values from Table 1. Note that RABBit invokes RACoon, which in turn calls $\alpha$-CROWN and eliminates verified executions (Line 7 in Algorithm 1). Hence, for a fair comparison, we also run $\alpha$-CROWN first for $\alpha, \beta$-CROWN and then run $\alpha, \beta$-CROWN only on the unverified indices. For all DNNs, RABBit consistently outperforms the SOTA BaB-based non-relational verifier $\alpha, \beta$-CROWN at all timestamps. This confirms that the improved precision shown in Table 1 is not dependent on the specific timeout value.

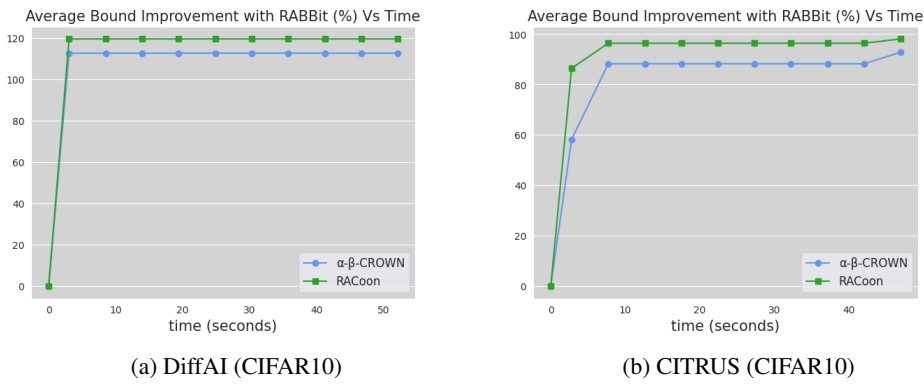

(a) DiffAI (CIFAR10)  (b) CITRUS (CIFAR10)

Figure 2: Timewise Analysis of Average % Improvement in $t^*$ with Strong Bounding (CIFAR10)

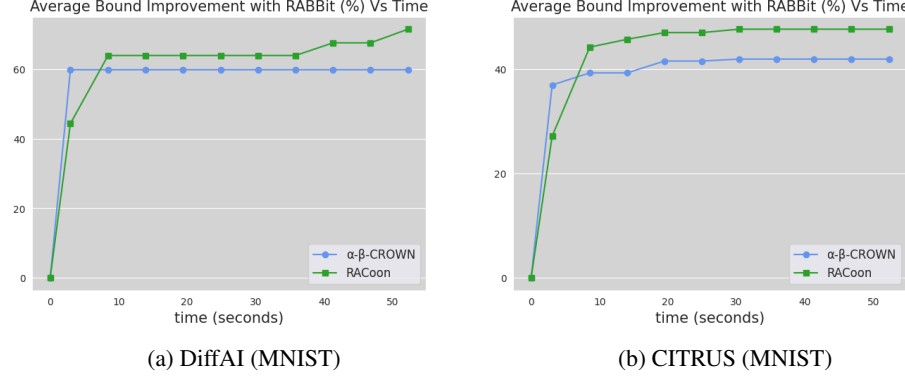

(a) DiffAI (MNIST)  (b) CITRUS (MNIST)

Figure 3: Timewise Analysis of Average % Improvement in $t^*$ with Strong Bounding (MNIST)

**Evaluating Bound Improvement:** In Figs 2 and 3, we present a timewise analysis of the improvement in $t^*$ with strong bounding over $\alpha, \beta$-CROWN and RACoon. For this experiment, we use DiffAI and CITRUS ConvSmall networks with epsilon values from Table 1. For each network and $\epsilon$, we select 30 executions at random and compute the percentage improvement in $t^*$ with strong bounding over RACoon and $\alpha, \beta$-CROWN. We also report the average improvement and 95% confidence intervals for all cases in Table 6 in Appendix G. The results demonstrate that the $t^*$ with strong bounding is significantly tighter compared to the bounds from the SOTA verifiers $\alpha, \beta$-CROWN and

RACoon at all timestamps. Furthermore, strong bounding improves $t^*$ on average by up to 108.7% for CIFAR10 networks and 57.7% for MNIST networks. These results highlight the importance of leveraging dependencies across executions during both branching and bounding to improve precision.

**Different $\epsilon$ and $k$ values:** Fig. 4 shows the results of RACoon, $\alpha, \beta$-CROWN, and RABBit for $k$-UAP verification of CIFAR10 ConvSmall DNNs for 5 different $\epsilon$ values and $k = 50$. RABBit outperforms RACoon and $\alpha, \beta$-CROWN for all evaluated $\epsilon$ values, notably improving the worst case $k$-UAP accuracy by up to 4.8%. Similarly, we analyze the performance of RACoon, $\alpha, \beta$-CROWN, and RABBit for $k$-UAP verification of CIFAR10 ConvSmall DNNs with different $k$ values. As presented in Fig. 5, for all $k$ values, RABBit is more precise than both baselines. Expectedly, the worst-case $k$-UAP accuracy for relational verifiers is higher with larger $k$ values as it is easier to prove the absence of a common perturbation with larger $k$.

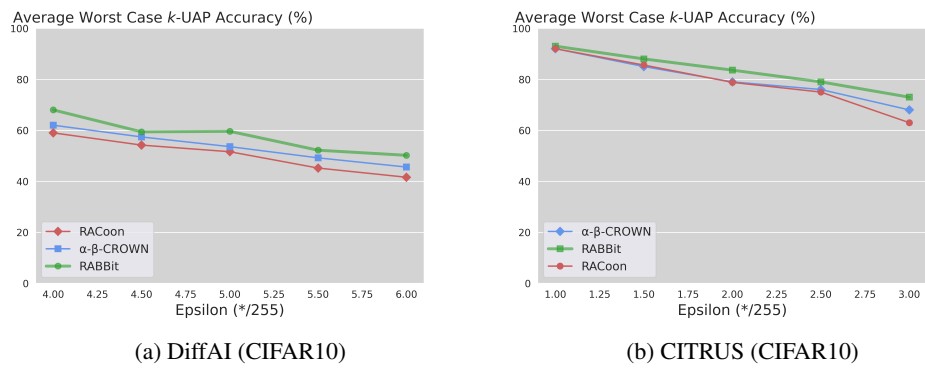

(a) DiffAI (CIFAR10)  (b) CITRUS (CIFAR10)

Figure 4: Average Worst Case $k$-UAP accuracy vs $\epsilon$ for CIFAR10 DNNs.

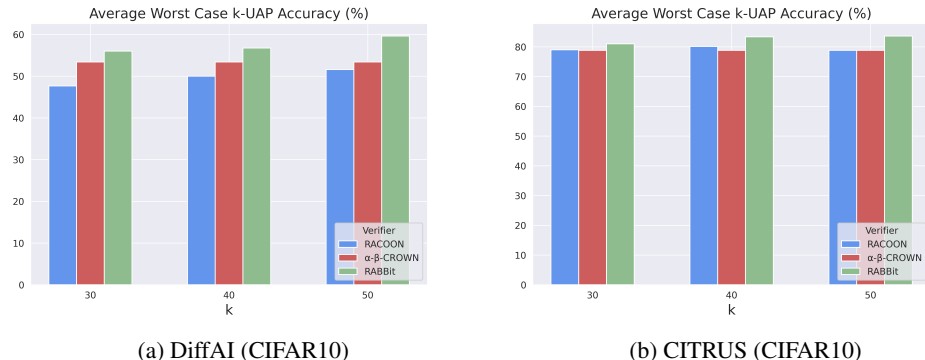

(a) DiffAI (CIFAR10)  (b) CITRUS (CIFAR10)

Figure 5: Average Worst Case $k$-UAP accuracy for different $k$ values for CIFAR10 ConvSmall DNNs.

## 7 Conclusion

We present RABBit, a general framework for improving the precision of relational verification of DNNs through BaB methods specifically designed to utilize dependencies across executions. Our experiments, on various DNN architectures, and training methods demonstrate that RABBit significantly outperforms both SOTA relational and non-relational verifiers for relational properties. Although we focus on the worst-case UAP accuracy and top-k accuracy RABBit can be extended to properties involving different DNNs, such as local equivalence of DNN pairs Paulsen et al. [2020] or properties defined over an ensemble of DNNs.

## Acknowledgement

We thank the anonymous reviewers for their insightful comments. This work was supported in part by NSF Grants No. CCF-2238079, CCF-2316233, CNS-2148583.

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

# A    Formal encoding of relational properties

## A.1    k-UAP verification

Given a set of $k$ points $\mathbf{X} = \{\mathbf{x_1}, ..., \mathbf{x_k}\}$ where for all $i \in [k]$, $\mathbf{x_i} \in \mathbb{R}^{n_0}$ and $\epsilon \in \mathbb{R}$ we can first define individual input constraints used to define $L_\infty$ input region for each execution $\forall i \in [k].\phi_{in}^i(\mathbf{x_i^*}) = \|\mathbf{x_i^*} - \mathbf{x_i}\|_\infty \le \epsilon$. We define $\Phi^\delta(\mathbf{x_1^*}, \ldots, \mathbf{x_k^*})$ as follows:

$$\Phi^\delta(\mathbf{x_1^*}, \ldots, \mathbf{x_k^*}) = \bigwedge_{(i,j \in [k]) \wedge (i<j)} (\mathbf{x_i^*} - \mathbf{x_j^*} = \mathbf{x_i} - \mathbf{x_j}) \tag{2}$$

Then, we have the input specification as $\Phi(\mathbf{x_1^*}, \ldots, \mathbf{x_k^*}) = \bigwedge_{i=1}^k \phi_{in}^i(\mathbf{x_i^*}) \wedge \Phi^\delta(\mathbf{x_1^*}, \ldots, \mathbf{x_k^*})$.

Next, we define $\Psi(\mathbf{x_1^*}, \ldots, \mathbf{x_k^*})$ as conjunction of $k$ clauses each defined by $\psi^i(\mathbf{y_i})$ where $\mathbf{y_i} = N(\mathbf{x_i^*})$. Now we define $\psi^i(\mathbf{y_i}) = \bigwedge_{j=1}^{n_l}(\mathbf{c_{i,j}}^T \mathbf{y_i} \ge 0)$ where $\mathbf{c_{i,j}} \in \mathbb{R}^{n_l}$ is defined as follows

$$\forall a \in [n_l].c_{i,j,a} = \begin{cases} 1 & \text{if } a \ne j \text{ and } a \text{ is the correct label for } \mathbf{y_i} \\ -1 & \text{if } a = j \text{ and } a \text{ is not the correct label for } \mathbf{y_i} \\ 0 & \text{otherwise} \end{cases} \tag{3}$$

In this case, the tuple of inputs $(\mathbf{x_1^*}, \ldots, \mathbf{x_k^*})$ satisfies the input specification $\Phi(\mathbf{x_1^*}, \ldots, \mathbf{x_k^*})$ iff for all $i \in [k]$, $\mathbf{x_i^*} = \mathbf{x_i} + \boldsymbol{\delta}$ where $\boldsymbol{\delta} \in \mathbb{R}^{n_0}$ and $\|\boldsymbol{\delta}\|_\infty \le \epsilon$. Hence, the relational property $(\Phi, \Psi)$ defined above verifies whether there is an adversarial perturbation $\boldsymbol{\delta} \in \mathbb{R}^{n_0}$ with $\|\boldsymbol{\delta}\|_\infty \le \epsilon$ that can misclassify **all** $k$ inputs. Next, we show the formulation for the worst-case UAP accuracy of the k-UAP verification problem as described in section 3. Let, for any $\boldsymbol{\delta} \in \mathbb{R}^{n_0}$ and $\|\boldsymbol{\delta}\|_\infty \le \epsilon$, $\mu(\delta)$ denotes the number of clauses ($\psi^i$) in $\Psi$ that are satisfied. Then $\mu(\delta)$ is defined as follows

$$z_i(\boldsymbol{\delta}) = \begin{cases} 1 & \psi^i(N(\mathbf{x_i} + \boldsymbol{\delta})) \text{ is } True \\ 0 & \text{otherwise} \end{cases} \tag{4}$$

$$\mu(\boldsymbol{\delta}) = \sum_{i=1}^k z_i(\boldsymbol{\delta}) \tag{5}$$

Since $\psi^i(N(\mathbf{x_i} + \boldsymbol{\delta}))$ is $True$ iff the perturbed input $\mathbf{x_i} + \boldsymbol{\delta}$ is correctly classified by $N$, for any $\boldsymbol{\delta} \in \mathbb{R}^{n_0}$ and $\|\boldsymbol{\delta}\|_\infty \le \epsilon$, $\mu(\boldsymbol{\delta})$ captures the number of correct classifications over the set of perturbed inputs $\{\mathbf{x_1} + \boldsymbol{\delta}, \ldots, \mathbf{x_k} + \boldsymbol{\delta}\}$. The worst-case k-UAP accuracy $\mathbf{M}_0(\Phi, \Psi)$ for $(\Phi, \Psi)$ is as follows

$$\mathbf{M}_0(\Phi, \Psi) = \min_{\boldsymbol{\delta} \in \mathbb{R}^{n_0}, \|\boldsymbol{\delta}\| \le \epsilon} \mu(\boldsymbol{\delta}) \tag{6}$$

# B    Details of strong bounding

For fixed linear approximations $\{(\mathbf{L}_1, b_1), \ldots (\mathbf{L}_n, b_n)\}$ corresponding to $n$ executions of $N$ if the optimal value $t^*$ of the following linear program $\ge 0$ then the $n$ executions do not have a common peturbation (from Theorem B.3. Banerjee and Singh [2024]).

$$min \ \ t \ \ \text{s.t.} \ \|\boldsymbol{\delta}\|_\infty \le \epsilon$$
$$\mathbf{L_i}^T(\mathbf{x_i} + \boldsymbol{\delta}) + b_i \le t \ \ \ \forall i \in [n] \tag{7}$$

Now in the first step, we compute the Lagrangian dual of the linear program from Eq. 7. The Lagrangian Dual is as follows where for all $i \in [n]$, $\lambda_i \ge 0$ are Lagrange multipliers.

$$\max_{0 \le \lambda_i} \min_{t \in \mathbb{R}, \|\boldsymbol{\delta}\|_\infty \le \epsilon} (1 - \sum_{i=1}^n \lambda_i) \times t + \sum_{i=1}^n \lambda_i \times \left(\mathbf{L}_i^T(\mathbf{x_i} + \boldsymbol{\delta}) + b_i\right)$$

We set the coefficient of the unbounded variable $t$ to 0 to avoid cases where $\min_{t \in \mathbb{R}, \|\boldsymbol{\delta}\|_\infty \le \epsilon} (1 - \sum_{i=1}^n \lambda_i) \times t + \sum_{i=1}^n \lambda_i \times \left(\mathbf{L}_i^T(\mathbf{x_i} + \boldsymbol{\delta}) + b_i\right) = -\infty$. This leads to the following Lagrangian Dual form

$$\max_{0 \le \lambda_i} \min_{\|\boldsymbol{\delta}\|_\infty \le \epsilon} \sum_{i=1}^n \lambda_i \times \left(\mathbf{L}_i^T(\mathbf{x_i} + \boldsymbol{\delta}) + b_i\right) \ \ \ \text{where} \sum_{i=1}^n \lambda_i = 1$$

Now for every subproblem, replacing the branching constraints by introducing dual variables $\boldsymbol{\beta}$ results in the parametric linear approximations of $N$ specified by $(\mathbf{L}_i(\boldsymbol{\alpha}_i, \boldsymbol{\beta}_i), b_i(\boldsymbol{\alpha}_i, \boldsymbol{\beta}_i))$ for each execution $i \in [n]$. Then the Lagrangian Dual with the parametric linear approximations $\{(\mathbf{L}_1(\boldsymbol{\alpha}_1, \boldsymbol{\beta}_1), b_1(\boldsymbol{\alpha}_1, \boldsymbol{\beta}_1)), \ldots, (\mathbf{L}_n(\boldsymbol{\alpha}_n, \boldsymbol{\beta}_n), b_n(\boldsymbol{\alpha}_n, \boldsymbol{\beta}_n))\}$ is as follows

$$\max_{0 \leq \lambda_i} \min_{\|\boldsymbol{\delta}\|_\infty \leq \epsilon} \sum_{i=1}^{n} \lambda_i \times \left(\mathbf{L}_i(\boldsymbol{\alpha}_i, \boldsymbol{\beta}_i)^T (\mathbf{x_i} + \boldsymbol{\delta}) + b_i(\boldsymbol{\alpha}_i, \boldsymbol{\beta}_i)\right) \quad \text{where } \sum_{i=1}^{n} \lambda_i = 1$$

**Theorem 4.1.** *If* $\min_{\mathcal{F}(S)} \max_{\boldsymbol{\alpha}_i, \boldsymbol{\beta}_i, \lambda_i} -\epsilon \times \|\sum_{i \in S} \lambda_i \times \mathbf{L}_i(\boldsymbol{\alpha}_i, \boldsymbol{\beta}_i)\|_1 + \sum_{i \in S} \lambda_i \times a_i(\boldsymbol{\alpha}_i, \boldsymbol{\beta}_i) \geq 0$ *then executions in* $S$ *do not have a common perturbation* $\boldsymbol{\delta} \in \mathbb{R}^{n_0}$ *with* $\|\boldsymbol{\delta}\|_\infty \leq \epsilon$.

*Proof.* First, we show that $\min_{\|\boldsymbol{\delta}\|_\infty \leq \epsilon} \sum_{i=1}^{n} \lambda_i \times \left(\mathbf{L}_i(\boldsymbol{\alpha}_i, \boldsymbol{\beta}_i)^T (\mathbf{x_i} + \boldsymbol{\delta}) + b_i(\boldsymbol{\alpha}_i, \boldsymbol{\beta}_i)\right) = -\epsilon \times \|\sum_{i=1}^{n} \lambda_i \times \mathbf{L}_i(\boldsymbol{\alpha}_i, \boldsymbol{\beta}_i)\|_1 + \sum_{i=1}^{n} \lambda_i \times a_i(\boldsymbol{\alpha}_i, \boldsymbol{\beta}_i)$.

$$\min_{\|\boldsymbol{\delta}\|_\infty \leq \epsilon} \sum_{i=1}^{n} \lambda_i \times \left(\mathbf{L}_i(\boldsymbol{\alpha}_i, \boldsymbol{\beta}_i)^T (\mathbf{x_i} + \boldsymbol{\delta}) + b_i(\boldsymbol{\alpha}_i, \boldsymbol{\beta}_i)\right)$$

$$= \min_{\|\boldsymbol{\delta}\|_\infty \leq \epsilon} \sum_{i=1}^{n} \lambda_i \times \mathbf{L}_i(\boldsymbol{\alpha}_i, \boldsymbol{\beta}_i)^T (\boldsymbol{\delta}) + \sum_{i=1}^{n} \lambda_i \times \left(b_i(\boldsymbol{\alpha}_i, \boldsymbol{\beta}_i) + \mathbf{L}_i(\boldsymbol{\alpha}_i, \boldsymbol{\beta}_i)^T \mathbf{x_i}\right)$$

$$= \sum_{i=1}^{n} \lambda_i \times a_i(\boldsymbol{\alpha}_i, \boldsymbol{\beta}_i) + \min_{\|\boldsymbol{\delta}\|_\infty \leq \epsilon} \sum_{i=1}^{n} \lambda_i \times \mathbf{L}_i(\boldsymbol{\alpha}_i, \boldsymbol{\beta}_i)^T (\boldsymbol{\delta})$$

$$= \sum_{i=1}^{n} \lambda_i \times a_i(\boldsymbol{\alpha}_i, \boldsymbol{\beta}_i) - \epsilon \times \|\sum_{i=1}^{n} \lambda_i \times \mathbf{L}_i(\boldsymbol{\alpha}_i, \boldsymbol{\beta}_i)\|_1 \quad \text{Using Hölder's Inequality} \qquad (8)$$

For fixed $\boldsymbol{\alpha}_i, \boldsymbol{\beta}_i$, the optimal solution of the LP in Eq. 7 and subsequently of the Lagrangian gives us

$$\max_{0 \leq \lambda_i} \min_{\|\boldsymbol{\delta}\|_\infty \leq \epsilon} \sum_{i=1}^{n} \lambda_i \times \left(\mathbf{L}_i(\boldsymbol{\alpha}_i, \boldsymbol{\beta}_i)^T (\mathbf{x_i} + \boldsymbol{\delta}) + b_i(\boldsymbol{\alpha}_i, \boldsymbol{\beta}_i)\right)$$

$$= \min_{\|\boldsymbol{\delta}\|_\infty \leq \epsilon} \max_{1 \leq i \leq n} \left(\mathbf{L}_i(\boldsymbol{\alpha}_i, \boldsymbol{\beta}_i)^T (\mathbf{x_i} + \boldsymbol{\delta}) + b_i(\boldsymbol{\alpha}_i, \boldsymbol{\beta}_i)\right) \quad \text{provided } \sum_{i=1}^{n} \lambda_i = 1 \qquad (9)$$

For each subproblem, for all $\boldsymbol{\alpha}_i, \boldsymbol{\beta}_i$

$$\min_{\|\boldsymbol{\delta}\|_\infty \leq \epsilon} \max_{1 \leq i \leq n} \mathbf{c_i}^T N(\mathbf{x_i} + \boldsymbol{\delta}) \geq \min_{\|\boldsymbol{\delta}\|_\infty \leq \epsilon} \max_{1 \leq i \leq n} \left(\mathbf{L}_i(\boldsymbol{\alpha}_i, \boldsymbol{\beta}_i)^T (\mathbf{x_i} + \boldsymbol{\delta}) + b_i(\boldsymbol{\alpha}_i, \boldsymbol{\beta}_i)\right)$$

Hence,

$$\min_{\|\boldsymbol{\delta}\|_\infty \leq \epsilon} \max_{1 \leq i \leq n} \mathbf{c_i}^T N(\mathbf{x_i} + \boldsymbol{\delta})$$

$$\geq \max_{\boldsymbol{\alpha}_i, \boldsymbol{\beta}_i} \min_{\|\boldsymbol{\delta}\|_\infty \leq \epsilon} \max_{1 \leq i \leq n} \left(\mathbf{L}_i(\boldsymbol{\alpha}_i, \boldsymbol{\beta}_i)^T (\mathbf{x_i} + \boldsymbol{\delta}) + b_i(\boldsymbol{\alpha}_i, \boldsymbol{\beta}_i)\right)$$

$$\geq \max_{\boldsymbol{\alpha}_i, \boldsymbol{\beta}_i} \max_{0 \leq \lambda_i} \min_{\|\boldsymbol{\delta}\|_\infty \leq \epsilon} \sum_{i=1}^{n} \lambda_i \times \left(\mathbf{L}_i(\boldsymbol{\alpha}_i, \boldsymbol{\beta}_i)^T (\mathbf{x_i} + \boldsymbol{\delta}) + b_i(\boldsymbol{\alpha}_i, \boldsymbol{\beta}_i)\right) \quad \text{where } \sum_{i=1}^{n} \lambda_i = 1 \text{ from Eq. 9}$$

$$\geq \max_{\boldsymbol{\alpha}_i, \boldsymbol{\beta}_i, 0 \leq \lambda_i} \sum_{i=1}^{n} \lambda_i \times a_i(\boldsymbol{\alpha}_i, \boldsymbol{\beta}_i) - \epsilon \times \|\sum_{i=1}^{n} \lambda_i \times \mathbf{L}_i(\boldsymbol{\alpha}_i, \boldsymbol{\beta}_i)\|_1 \quad \text{From Eq. 8} \qquad (10)$$

Finally, if $\min_{\mathcal{F}(S)} \max_{\boldsymbol{\alpha}_i, \boldsymbol{\beta}_i, \lambda_i} -\epsilon \times \|\sum_{i \in S} \lambda_i \times \mathbf{L}_i(\boldsymbol{\alpha}_i, \boldsymbol{\beta}_i)\|_1 + \sum_{i \in S} \lambda_i \times a_i(\boldsymbol{\alpha}_i, \boldsymbol{\beta}_i) \geq 0$ then,

$$\min_{\|\boldsymbol{\delta}\|_\infty \leq \epsilon} \max_{1 \leq i \leq n} \mathbf{c_i}^T N(\mathbf{x_i} + \boldsymbol{\delta}) \geq 0 \quad \text{for all subproblems in } \mathcal{F}(S) \text{ using Eq. 10}$$

Since, for all subproblems $\min_{\|\boldsymbol{\delta}\|_\infty \leq \epsilon} \max_{1 \leq i \leq n} \mathbf{c_i}^T N(\mathbf{x_i} + \boldsymbol{\delta}) \geq 0$, $\bigvee_{i=1}^{n} \psi^i(N(\mathbf{x_i} + \boldsymbol{\delta}))$ holds for all $\boldsymbol{\delta} \in \mathbb{R}^{n_0}$ and $\|\boldsymbol{\delta}\|_\infty \leq \epsilon$ i.e. there does not exist any common perturbation. $\square$

# C  Details of strong branching

**Theorem 4.2.** *For any $\boldsymbol{\alpha}, \boldsymbol{\beta}$, if $\mathbf{L}(\boldsymbol{\alpha}, \boldsymbol{\beta}) \in \mathbb{R}^{n_0}$ and $b(\boldsymbol{\alpha}, \boldsymbol{\beta}) \in \mathbb{R}$ then $\min_{\|\boldsymbol{\delta}\|_\infty \leq \epsilon}(\mathbf{L}(\boldsymbol{\alpha}, \boldsymbol{\beta}) - \mathbf{L}_t)^T(\mathbf{x} + \boldsymbol{\delta}) + b(\boldsymbol{\alpha}, \boldsymbol{\beta}) = -\epsilon \times \|\mathbf{L}(\boldsymbol{\alpha}, \boldsymbol{\beta}) - \mathbf{L}_t\|_1 + (\mathbf{L}(\boldsymbol{\alpha}, \boldsymbol{\beta}) - \mathbf{L}_t)^T\mathbf{x} + b(\boldsymbol{\alpha}, \boldsymbol{\beta}).$*

*Proof.*

$$
\begin{aligned}
&\min_{\|\boldsymbol{\delta}\|_\infty \leq \epsilon} (\mathbf{L}(\boldsymbol{\alpha}, \boldsymbol{\beta}) - \mathbf{L}_t)^T(\mathbf{x} + \boldsymbol{\delta}) + b(\boldsymbol{\alpha}, \boldsymbol{\beta}) \\
&= \min_{\|\boldsymbol{\delta}\|_\infty \leq \epsilon} (\mathbf{L}(\boldsymbol{\alpha}, \boldsymbol{\beta}) - \mathbf{L}_t)^T\boldsymbol{\delta} + b(\boldsymbol{\alpha}, \boldsymbol{\beta}) + (\mathbf{L}(\boldsymbol{\alpha}, \boldsymbol{\beta}) - \mathbf{L}_t)^T\mathbf{x} \\
&= b(\boldsymbol{\alpha}, \boldsymbol{\beta}) + (\mathbf{L}(\boldsymbol{\alpha}, \boldsymbol{\beta}) - \mathbf{L}_t)^T\mathbf{x} + \min_{\|\boldsymbol{\delta}\|_\infty \leq \epsilon} (\mathbf{L}(\boldsymbol{\alpha}, \boldsymbol{\beta}) - \mathbf{L}_t)^T\boldsymbol{\delta} \\
&= b(\boldsymbol{\alpha}, \boldsymbol{\beta}) + (\mathbf{L}(\boldsymbol{\alpha}, \boldsymbol{\beta}) - \mathbf{L}_t)^T\mathbf{x} - \epsilon \times \|(\mathbf{L}(\boldsymbol{\alpha}, \boldsymbol{\beta}) - \mathbf{L}_t)\|_1 \quad \text{Using Hölder's Inequality}
\end{aligned}
$$

$\square$

## C.1  Projected gradient ascent

For each $\boldsymbol{\alpha_i}, \boldsymbol{\beta_i}$, after each step of gradient ascent (for maximization problem), we clip $\boldsymbol{\alpha_i}, \boldsymbol{\beta_i}$ values to the corresponding ranges $[\boldsymbol{l_i^\alpha}, \boldsymbol{u_i^\alpha}]$ $[\boldsymbol{l_i^\beta}, \boldsymbol{u_i^\beta}]$ respectively. This is similar to the approach used in the SOTA non-relational bound refinement $\alpha, \beta$-CROWN Wang et al. [2021]. Since $\lambda_i \in [0, 1]$ and $\sum_{i=1}^k \lambda_i = 1$ we replace $\lambda_i = \frac{sigmoid(x_i)}{\sum_{i=1}^k sigmoid(x_i)}$ where $x_i \in \mathbb{R}$. For any values of $(x_1, \ldots, x_k) \in \mathbb{R}^k$ the corresponding $(\lambda_1, \ldots, \lambda_k)$ satisfy $\lambda_i \in [0, 1]$ and $\sum_{i=1}^k \lambda_i = 1$. We then apply gradient ascent (for maximization problem) on $(x_1, \ldots, x_k)$ without any constraints.

# D    DNN Architectures

## D.1    DNN Architectures:

Table 2: DNN Architecture Details

| Dataset | Model | Type | Train | # Layers | # Params |
|---------|-------|------|-------|----------|----------|
| MNIST | ConvSmall | Conv | Standard | 4 | 80k |
| | ConvSmall | Conv | DiffAI | 4 | 80k |
| | ConvSmall | Conv | SABR | 4 | 80k |
| | ConvSmall | Conv | CITRUS | 4 | 80k |
| | ConvBig | Conv | DiffAI | 7 | 1.8M |
| CIFAR10 | ConvSmall | Conv | Standard | 4 | 80k |
| | ConvSmall | Conv | DiffAI | 4 | 80k |
| | ConvSmall | Conv | SABR | 4 | 80k |
| | ConvSmall | Conv | CITRUS | 4 | 80k |
| | ConvBig | Conv | DiffAI | 7 | 2.5M |

## D.2    Accuracies for Evaluated DNNs:

Table 3: Standard top-1 accuracy for evaluated DNNs

| Dataset | Model | Train | Accuracy (%) |
|---------|-------|-------|--------------|
| CIFAR10 | ConvSmall | Standard | 62.9 |
| | ConvSmall | DiffAI | 45.9 |
| | ConvSmall | SABR | 63.3 |
| | ConvSmall | CITRUS | 63.9 |
| | ConvBig | DiffAI | 53.8 |
| MNIST | ConvSmall | Standard | 97.7 |
| | ConvSmall | DiffAI | 96.8 |
| | ConvSmall | SABR | 97.9 |
| | ConvSmall | CITRUS | 98.5 |

# E Comparison of RABBit with RaVeN

Table 4: Comparison of RABBit vs RaVeN and RACoon

| Dataset | Network | Training Method | $\epsilon$ | RaVeN | RACoon | RABBit |
|---------|---------|-----------------|------------|-------|--------|--------|
| MNIST | ConvSmall | DiffAI | 0.13 | 66.0 | 63.0 | 82.0 (+19.0) |
| MNIST | ConvSmall | SABR | 0.15 | 51.0 | 50.0 | 66.0 (+15.0) |
| MNIST | ConvSmall | CITRUS | 0.14 | 82.0 | 80.0 | 89.0 (+7.0) |
| CIFAR10 | ConvSmall | SABR | 4.0/255 | 48.0 | 48.0 | 55.0 (+7.0) |
| CIFAR10 | ConvSmall | CITRUS | 2.0/255 | 79.0 | 78.0 | 82.0 (+3.0) |

# F RABBit Efficacy Analysis for top-k accuracy

Table 5: Verified top-2 accuracy for RABBit vs baselines

| Dataset | Network | Training Method | $\epsilon$ | $\alpha$-CROWN | | RACoon | | $\alpha, \beta$-CROWN | | RABBit | |
|---------|---------|-----------------|------------|----------------|----------------|----------------|----------------|----------------|----------------|----------------|----------------|
| | | | | Avg. Acc. (%) | Avg. Time (sec.) | Avg. Acc. (%) | Avg. Time (sec.) | Avg. Acc. (%) | Avg. Time (sec.) | Avg. Acc. (%) | Avg. Time (sec.) |
| CIFAR10 | ConvSmall | DiffAI | 5/255 | 74.0 | 4.52 | 75.0 | 4.87 | 75.0 | 20.47 | 78.0 (+3.0) | 24.27 |
| MNIST | ConvSmall | DiffAI | 0.13 | 84.0 | 1.20 | 84.0 | 1.42 | 89.0 | 11.03 | 91.0 (+2.0) | 13.43 |

# G Average Improvement in $t^*$ with Strong Branching

Table 6: Average Improvement in $t^*$ with Strong Bounding

| Dataset | Network Structure | Training Method | Perturbation Bound ($\epsilon$) | RACoon | | $\alpha, \beta$-CROWN | |
|---------|-------------------|-----------------|----------------------------------|-----------------------|----------------|-----------------------|----------------|
| | | | | Avg. Improvement (%) | 95% CI | Avg. Improvement (%) | 95% CI |
| CIFAR | ConvSmall | DiffAI | 5/255 | 108.7 | [93.9, 126.1] | 102.5 | [92.7, 115.4] |
| | ConvSmall | CITRUS | 2/255 | 77.9 | [75.3, 81.6] | 86.9 | [86.2, 88.1] |
| MNIST | ConvSmall | DiffAI | 5/255 | 57.7 | [55.5, 60.2] | 54.4 | [53.0, 56.0] |
| | ConvSmall | CITRUS | 2/255 | 40.8 | [39.8, 41.9] | 37.1 | [36.4, 37.8] |

