# OpenReview forum: "Relational Verification Leaps Forward with RABBit"
_NeurIPS.cc/2024/Conference — NeurIPS 2024 poster_

### Official Review · Reviewer_aDbJ · 2024-06-18

**Soundness:** 2
**Presentation:** 1
**Contribution:** 3
**Rating:** 3
**Confidence:** 4

**Summary:**

The paper proposes bound-tightening techniques for verifying the absence of universal adversarial perturbations for a neural network. The tightened bounds are leveraged in a MILP encoding to perform verification.

**Strengths:**

- The paper addresses an interesting problem: verifying the absence of universal adversarial perturbations.
- It introduces a novel algorithm that combines ideas from previous neural network verifiers.
- It also introduces a novel bound computation approach that optimises only the offsets of the linear bounds (Strong Branching).
- The proofs of the theoretical results are valid.
- The experiments could demonstrate that the proposed verifier (RABBit) outperforms the baselines but I have some concerns regarding the timeouts (see Questions).
- The authors provide code in the supplementary material (but see Weaknesses)

**Weaknesses:**

### Baselines
The non-relational verifier baselines CROWN [Zhang et al. 2018], $\alpha$-CROWN [Xu et al., 2021], $\alpha,\beta$-CROWN verifier [Wang et al., 2021b] are no longer state of the art. These baselines were outperformed by GCP-CROWN [Zhang et al., 2022] and MN-BaB [Ferrari et al., 2022]. To justify the claims on outperforming the non-relational SOTA, the paper should compare against GCP-CROWN and/or MN-BaB.

### Sensitivity of Results
For the "Effectiveness of RABBit" and the "Time vs UAP Accuracy Analysis" experiments, the authors use ten randomly sampled problem instances but only report the average results. It remains unclear how sensitive the results are to the sampling.

### Presentation
- The paper is not sufficiently self-contained. In particular, it does not provide sufficient details on RACoon which it builds upon.
- I find the structure of the paper confusing. I only understood Section 4 after reading Section 5. I think presenting Algorithm 1 before Strong Bounding and Strong Branching would be more clear.
- Important mathematical details are spread throughout the paper. For example, the maximisation in Theorem 4.1 has additional constraints on $\lambda_i$ that are mentioned in lines 206-207 and also constraints on $\boldsymbol{\alpha}$ and $\boldsymbol{\beta}$ that are only mentioned in lines 128 and 135.

### Code
The code included in the supplementary material is insufficiently documented (what do I have to do to reproduce the experiments? What is "raven"?) and buggy:
- The requirements can not be installed as specified in the README (autoLiRPA 0.4 is not available on PyPI at the time of writing; Can not install the specified PyTorch versions without an `--extra-index-url`).
- The test command that is described in the README fails due to a syntax error in file "raven/src/util.py" at line 247 `(root'', train=train, download=True, transform=transform_test)`.
- The supplementary material does not include the models used in the experiments, describe how to obtain them, or indicate where to place them.


[Brix et al., 2023]: Christopher Brix, Stanley Bak, Changliu Liu, Taylor T. Johnson: The Fourth International Verification of Neural Networks Competition (VNN-COMP 2023): Summary and Results. CoRR abs/2312.16760 (2023)
[Zhang et al., 2022]: Huan Zhang, Shiqi Wang, Kaidi Xu, Linyi Li, Bo Li, Suman Jana, Cho-Jui Hsieh, J. Zico Kolter: General Cutting Planes for Bound-Propagation-Based Neural Network Verification. NeurIPS 2022.

**Questions:**

### Questions
1. I am confused by the statement "Each MILP instance gets a timeout of 5 minutes" (line 351). With $k=50$ and $k_t=20$ for MNIST, 5 minutes per MILP already add up to 100 minutes (2x the timeout) at the end of the first for loop (line 19), before Algorithm 1 considers the timeout in line 20. What are the actual runtimes of RABBit?
2. Table 1: How were the perturbation radii chosen?
3. The worst-case UAP certified by RABBit in Table 1 is higher than the standard accuracy (Table 3) for several networks (CIFAR10: ConvSmall+DiffAI, ConvSmall+SABR, ConvSmall+CITRUS, ConvBig+DiffAI, MNIST: all networks). In my understanding, UAP should always be lower than standard accuracy (UAP = "the rate at which N misclassifies when the same adversarial perturbation is applied to all inputs from the distribution", Lines 153-154). What do the reported UAPs actually indicate?
4. Besides the average UAPs "Effectiveness of RABBit" and the "Time vs UAP Accuracy Analysis" experiments, what are the standard deviations, minima, maxima, and 25%, 50%, and 75% quartiles?

### Suggestions and Typos
- The abbreviation SOTA is relatively well known, but I would still suggest introducing it.
- Line 14: space between "diagnosis" and Amato et al.
- Line 14 and throughout: Use citep instead of citet.
- Line 15: What does it mean to "understand [...] their reliability"?
- Line 17: The references Potdevin et al. [2019] and Wu et al. [2023b] seem out of place here since they do not introduce adversarial attacks.
- Line 18: Sotoudeh and Thakur [2020] do not study adversarial attacks.
- Lines 23-26: The discussion in these lines is paraphrasing the discussion in the introduction of Banerjee and Singh [2024]. You should add a citation for this, for example: "As discussed by Banerjee and Singh [2024], recent studies [Li et al, 2019a] emphasize ..."
- Line 52: with *a* cross-executional bound method
- Related Work: I suggest adding a discussion of certified adversarial training approaches, such as [Balauca et al., 2024; De Palma, 2024] since this field is closely related.
- Line 68: prove *a* property
- Line 104: add a justification or reference for the NP-hardness of NN verification.
- Lines 127-128: add a citation for parametric linear relaxations
- Line 132: $\alpha,\!\beta$-CROWN *converts*
- Line 136: and *refines* the parameters
- Line 136: last $\alpha,\!\beta$ also bold?
- Line 168: for *a* $k$-UAP problem
- Lines 169-170: bold $\boldsymbol{\delta}$ as in the Preliminaries Section?
- Line 172: highlight*ing*
- Line 185: Then *if* the optimal value $t^\ast \geq 0$ this proves the absence ...
- Lines 185-187: It does not become very clear here that $||\delta||_\infty \leq \epsilon$ and the following equations are constraints to the maximum in line 185. I strongly suggest putting both the max and the constraints in an equation environment. The same also holds for other maximisations and minimisations.
- Line 191: In the remainder of the paper, the number of executions is $k$, not $n$.
- Line 192: *the* product DNN
- Line 195: adapt or adopt?
- Line 197: add a description of what the BaBSR score estimates.
- Line 216: have *a* common perturbation.
- Line 224: *exploring* $\frac{m}{n}$ subproblem*s*.
- Line 225: absence of *a* common perturbation.
- Line 236: compute *a* valid lower bound.
- Line 270: *Algo*.
- Line 310: Then finding (no *the*)
- Line 311: $min_{||\delta||_\infty \leq \epsilon}$?
- Line 317: denote *the* set
- Figure 2: the title and axis descriptions are barely legible.
- Line 413: invalid DOI
- Line 416: Where published?
- Line 422: "jan", the other references don't have a month
- Line 448: there is a published version at ICLR 2015 for this article
- Line 495: Conference missing.
- Lines 510-517: Duplicate entry
- Line 529: Where published?
- Bibliography: incoherent links/no links, sometimes DOI and URL.
- Line 602: Projected gradient *ascent*
- Table 2: What are the full architectures or where can I find them?

[Balauca et al., 2024]: Stefan Balauca, Mark Niklas Müller, Yuhao Mao, Maximilian Baader, Marc Fischer, Martin T. Vechev:
Overcoming the Paradox of Certified Training with Gaussian Smoothing. CoRR abs/2403.07095 (2024)
[De Palma et al., 2024]: Alessandro De Palma, Rudy R Bunel, Krishnamurthy Dj Dvijotham, M. Pawan Kumar, Robert Stanforth, Alessio Lomuscio: Expressive Losses for Verified Robustness via Convex Combinations. ICRL 2024. https://openreview.net/forum?id=mzyZ4wzKlM

**Limitations:**

The authors adequately discuss limitations and social impact.

---

> ### Author Rebuttal · Authors · 2024-08-07
>
> **Q1: Comparison with SOTA non-relational baseline MNBaB.**
>
> **R1:** We compared the performance of RABBit with the proposed baseline MNBaB across all ConvSmall CIFAR10 networks listed in Table 1 of the paper. The comparison used the same $\epsilon$ values, hardware, and timeout values as mentioned in Section 6.1. While MNBaB outperforms $\alpha,\beta$-CROWN on 3 of the 4 networks, it remains significantly less precise than RABBit, as shown in Table 4. Runtime comparisons are presented in Table 5. For each property, the runtime of a verifier is the timestamp when the verifier first achieves its maximum UAP accuracy within the time limit.
>
> Table 4 UAP accuracy of RABBit vs baselines including MNBaB
> | Training         |   $\epsilon$ | $\alpha$-CROWN| RACoon |   $\alpha,\beta$-CROWN |   MNBaB |  RABBit |
> |:-----------------|-------:|:-------------|------:|------:|------:|---------:|
> | Standard         | 1/255    | 45.4% | 45.4%  | 59.8%  | 55.0%  | **62.4%**  |
> | DiffAI         |  5/255  | 49.6%|  51.6%|  53.6%|  55.0% |**59.8%**   |
> | SABR   | 2/255   | 75.8% | 78.2% | 78.4% | 80.0% | **84.0%**  |
> | CITRUS | 2/255   | 76.0%| 78.8% | 79.0%  | 79.6% | **83.6%**  |
>
> Table 5 Average runtime of  RABBit vs baselines including MNBaB
> | Training         |   $\epsilon$ | $\alpha$-CROWN (sec.)| RACoon (sec.) |   $\alpha,\beta$-CROWN (sec.)|   MNBaB (sec.) |  RABBit (sec.) |
> |:-----------------|-------:|:-------------|------:|------:|------:|---------:|
> | Standard         | 1/255    | 3.3 | 542.8  | 1385.4  | 1168.92  | 2062.8  |
> | DiffAI         |  5/255  | 4.7|  710.3|  1419.3|  1142.43 | 2303.1   |
> | SABR   | 2/255   | 3.7 | 267.3 | 1107.5 | 800.32 | 1665.3  |
> | CITRUS | 2/255   | 3.7| 268 | 1083.4  | 778.96 | 1416.7  |
>
>
> **Q2: Analyse the sensitivity to randomly sampled problem instances. Report the standard deviations, minima, maxima, and 25%, 50%, and 75% quartiles?**
>
> **R2:** We report the standard deviations and quartiles of the worst-case UAP accuracy for the ConvSmall CITRUS [1] MNIST and CIFAR10 in Tables 6 and 7 respectively. We used the same experimental setup described in section 6.1 of the paper and the same 10 properties from Table 1 in the paper. For all the quartiles, RABBit significantly outperforms all the baselines. We will report the results for all networks in the revised version of the paper.
>
>   Table 6: Worst-case UAP accuracy statistics for CIFAR10 ConvSmall CITRUS [1]
> | Verifier         |   Mean | 95% CI       |   Std |   Min |   25% |   Median |   75% |   Max |
> |:-----------------|-------:|:-------------|------:|------:|------:|---------:|------:|------:|
> | CROWN            |   74.8 | [68.3, 81.3] |   8.7 |    58 |  72   |       76 |  81.5 |    86 |
> | alpha-CROWN      |   76   | [69.4, 82.6] |   8.8 |    58 |  74   |       77 |  81.5 |    88 |
> | RACoon           |   78.8 | [71.1, 86.5] |  10.2 |    58 |  76.5 |       80 |  85.5 |    **92** |
> | α-β-CROWN        |   79   | [72.6, 85.4] |   8.5 |    64 |  76   |       80 |  85   |    90 |
> | Strong Branching |   81.2 | [74.7, 87.7] |   8.6 |    62 |  **80**   |       83 |  86   |    **92** |
> | Strong Bounding  |   82.8 | [78.2, 87.4] |   6.1 |    72 |  **80**   |       84 |  86   |    **92** |
> | RABBit           |   **83.6** | [79.3, 87.9] |   5.6 |    **74** |  **80**   |       **85** |  **87.5** |    **92** |
>
>
> Table 7: Worst-case UAP accuracy statistics for CIFAR10 ConvSmall CITRUS [1]
> | Verifier         |   Mean | 95% CI       |   Std |   Min |   25% |   Median |   75% |   Max |
> |:-----------------|-------:|:-------------|------:|------:|------:|---------:|------:|------:|
> | CROWN            |   28.8 | [22.9, 34.7] |   7.8 |    22 |  22.5 |       24 |  36   |    44 |
> | alpha-CROWN      |   41.6 | [36.5, 46.7] |   6.8 |    32 |  36.5 |       40 |  45   |    54 |
> | RACoon           |   44.6 | [39.1, 50.1] |   7.3 |    36 |  38.5 |       42 |  49   |    58 |
> | α-β-CROWN        |   59.4 | [54.5, 64.3] |   6.5 |    50 |  54.5 |       57 |  65   |    **72** |
> | Strong Branching |   60.0 | [55.3, 64.7] |   6.2 |    52 |  54.5 |       57 |  65   |    **72** |
> | Strong Bounding  |   60.6 | [56.2, 65.8] |   6.3 |    52 |  55   |       59 |  **65.5** |    **72** |
> | RABBit|   **61.6** | [57.1, 66.1] |   5.9 |    **54** |  **55.5** |       **60** |  **65.5** |    **72** |
>
> [1] "Cross-Input Certified Training for Universal Perturbations", C. Xu, et. al., ECCV, 2024.
>
> **Q3: What are the actual runtimes of RABBit?**
>
> **R3:** Please refer to the answer to Q2 in the common response for the detailed runtime analysis of RABBit and the baselines.
> The overall time limit given to RABBit is $k=50$ minutes and the result of any MILP instance that was not completed in the given time limit ($k$ minutes) was not considered in the final result. Also, for efficiency, each MILP instance is formulated in either line 16 or line 25 of Algo. 1 is executed in a different thread in parallel to the BaB and does not block subsequent iterations of the loops (lines 10 and 20 of Algo. 1) In all cases, RABBit's runtime is always bounded by $k$ minutes.
>
> **Q4: Table 1: How were the perturbation radii chosen?**
>
> **R4:** The perturbation radii used in this work are based on the existing relational verifier RACoon [1]. Other non-relational verifiers like MNBaB [2] also use the same perturbation radii for networks with the same architecture. We also have evaluated the performance of RABBit with different $\epsilon$ values in Figure 3 of the paper.
>
> [1] "Relational DNN Verification With Cross Executional Bound Refinement", D. Banerjee, et. al., ICML, 2024.
>
> [2] "Complete Verification via Multi-Neuron Relaxation Guided Branch-and-Bound", C. Ferrari, et. al., ICLR, 2022.

---

> ### Author Response · Authors · 2024-08-07
>
> We are addressing the remaining concerns (after Q4) with additional clarification here.
>
> **Q5: The worst-case UAP certified by RABBit in Table 1 is higher than the standard accuracy (Table 3). What do the reported UAPs indicate?**
>
> **R5:** In Table 3 of the paper, we aim to report the standard accuracies (without $\epsilon$) of each of the networks from Table 1 on the CIFAR and MNIST datasets. We realized that we miscalculated the standard accuracies for the MNIST networks and accidentally included the $\epsilon$ column.  We have corrected Table 3 from the paper and updated it below in the following table, Table 8.
>
> Table 8: Standard accuracy for evaluated DNNs
> | Dataset | Model      | Train    | Accuracy (%) |
> |---------|------------|----------|--------------|
> |   CIFAR10      | ConvSmall  | Standard | 62.9         |
> | CIFAR10 | ConvSmall  | DiffAI   | 45.9         |
> | CIFAR10 | ConvSmall  | SABR     | 63.3         |
> |   CIFAR10      | ConvSmall  | CITRUS   | 63.9         |
> |   CIFAR10      | ConvBig    | DiffAI   | 53.8         |
> |    CIFAR10     | ResNet-2B  | Standard | 67.5         |
> |     MNIST    | ConvSmall  | Standard | 97.7         |
> |     MNIST    | ConvSmall  | DiffAI   | 96.8         |
> | MNIST   | ConvSmall  | SABR     | 97.9         |
> |    MNIST     | ConvSmall  | CITRUS   | 98.5         |
> |      MNIST   | ConvBig    | DiffAI   | 91.8         |
>
> As described in line 357, for each network, similar to the relational verifier RACoon [1], we filter out the images that were misclassified by the network and do not consider them for computing UAP accuracy. Thus, the reported UAP accuracy in Table 1 of the paper is computed on the **correctly** classified images. Overall, the UAP accuracy of the network (that does not filter out misclassified images) will be the standard accuracy $\times$ UAP accuracy from Table 1. We will clarify this in the revised version and report standard accuracy $\times$ UAP accuracy as well.
>
> [1] "Relational DNN Verification With Cross Executional Bound Refinement", D. Banerjee, et. al., ICML, 2024.
>
> **Q6: Code is insufficiently documented and buggy.**\
> **R6:** Thank you for pointing this out. We have fixed the reported issues and shared an anonymized repository with the Area Chair.
>
>
> **Q7: The paper is not sufficiently self-contained. In particular, it does not provide sufficient details on RACoon which it builds upon.**\
> **R7:** We will add the necessary background for RACoon in the revised version of the paper.
>
> **Q8: I find the structure of the paper confusing. I only understood Section 4 after reading Section 5. I think presenting Algorithm 1 before Strong Bounding and Strong Branching would be more clear.**
>
> **R8:** Thanks for the suggestion. We will update it in the revised version of the paper.
>
> **Q9: Add a description of what the BaBSR score estimates and of certified adversarial training approaches.**
>
> **R9:** Thanks for pointing this out. We will add a detailed description of these topics in the revised version of the paper.
>
> **Q10: Table 2: What are the full architectures or where can I find them?**
>
> **R10:** As mentioned in line 341 of the paper, the ConvSmall and ConvBig architectures are taken from the ERAN repository [1] and the ResNet-2B architecture is from the $\alpha,\beta$-CROWN repository [2].
>
> [1] https://github.com/eth-sri/eran
>
> [2] https://github.com/Verified-Intelligence/alpha-beta-CROWN
>
> **Q11: Bibliography Issues and Typos.**\
> **R11:** Thanks for pointing out the bibliography issues and typos. We will correct all mistakes with citations, spelling, grammar, and figure sizes in the revised version.

---

> > ### Comment · Reviewer_aDbJ · 2024-08-09
> >
> > Thank you for your answer, which addresses some of my concerns. I have a followup question:
> >
> > > R3: Also, for efficiency, each MILP instance is formulated in either line 16 or line 25 of Algo. 1 is executed in a different thread in parallel to the BaB and does not block subsequent iterations of the loops (lines 10 and 20 of Algo. 1) In all cases, RABBit's runtime is always bounded by $k$ minutes.
> >
> > As far as I can see, this wasn't discussed in the paper. Does RABBit use multiple threads that run on a single CPU core or does it leverage multiple CPU cores?

---

> > > ### Author Response · Authors · 2024-08-10
> > >
> > > Dear reviewer aDbJ,
> > >
> > > Thanks for your response. We are happy that our response has resolved some of your concerns.
> > >
> > > **Q1: Does RABBit use multiple threads that run on a single CPU core or does it leverage multiple CPU cores?**
> > >
> > > **R1:** RABBit and all the baselines use Gurobi V11.0 for MILP optimization (as mentioned in line 345 of the paper). By default, Gurobi utilizes multiple cores depending on their availability [1]. We applied the default Gurobi settings for RABBit and all the baselines.
> > > RABBit, similar to all the baselines, uses a single GPU for BaB-based methods and utilizes the CPU for all MILP tasks. The hardware details can be found in lines 348-349 of the paper. RABBit initiates MILP optimization in a new thread to avoid blocking the GPU-based BaB methods while using the same CPU resources available to the other baselines. The CPU utilization is determined by Gurobi's default settings, which are consistent across all baselines.
> > >
> > > Moreover, currently in RABBit, we issue a new MILP everytime a new constraints is added (line 16 and line 25 of Algo. 1). All these intermediate calls can be replaced with a single MILP with all avaialble constraints placed before $(T_{total} - T_{MILP})$. This eliminates the need to invoke the MILP solver in a new thread and reduces the number of MILP calls in RABBit to just one. The final results using this method are comparable to those presented in Table 1 of the paper.
> > >
> > > Table 9 UAP accuracy of RABBit with only 1 MILP call
> > > | Training         |   $\epsilon$ | $\alpha$-CROWN| RACoon |   $\alpha,\beta$-CROWN |   MNBaB |  RABBit |
> > > |:-----------------|-------:|:-------------|------:|------:|------:|---------:|
> > > | Standard         | 1/255    | 45.4% | 45.4%  | 59.8%  | 55.0%  | **62.4%**  |
> > > | DiffAI         |  5/255  | 49.6%|  51.6%|  53.6%|  55.0% |**59.8%**   |
> > > | SABR   | 2/255   | 75.8% | 78.2% | 78.4% | 80.0% | **84.0%**  |
> > > | CITRUS | 2/255   | 76.0%| 78.8% | 79.0%  | 79.6% | **83.6%**  |
> > >
> > > [1] https://support.gurobi.com/hc/en-us/community/posts/4408537958289-CPU-utilization

---

> > > > ### Comment · Reviewer_aDbJ · 2024-08-12
> > > > **Final Review Rating**
> > > >
> > > > I thank the authors for the additional experimental details. The rebuttal addressed many weaknesses of the paper, particularly regarding runtimes, and the authors eventually provided executable code. My main concerns were the baselines, the sensitivity of the results, the presentation, and the code.
> > > >
> > > > - For the baselines, the authors provided a comparison with MN-BaB, which is a large step towards substantiating their claim of outperforming non-relational SOTA neural network verifiers. However, GPC-CROWN [Zhang et al., 2022] is unarguably *the* state-of-the-art verifier [Brix et al., 2023; Müller et al., 2022], which mandates that the authors compare against it to substantiate their claim of outperforming SOTA in the paper.
> > > >
> > > > - Regarding the sensitivity of the results, the authors have reported detailed summary statistics in the rebuttal. This is great. However, the numbers show that their results are quite sensitive to the sampled properties. This means that a faithful evaluation requires more than the ten properties used by the authors and a careful sensitivity analysis.
> > > >
> > > > - For the presentation, the authors have acknowledged the issues and plan to correct them. However, in my opinion, the paper requires a major revision, which then has to go through another reviewing process. This particularly concerns the self-containedness of the paper. Additionally, the discussion with the authors has brought up more issues with the presentation: As was discussed, Algorithm 1 does not properly reflect the actual implementation of RABBit. This leads to inconsistencies, for example, with regard to the presented timeout mechanism and the actually deployed timeout mechanism.
> > > >
> > > > - Finally, the authors now provided executable code, which is a large improvement. However, the code is still insufficiently documented and, therefore, does not allow for adequate reproducibility.
> > > >
> > > > Overall, the authors have made large strides towards improving the paper. However, I still think that the paper is not yet ready for publication and the major revisions required mandate another reviewing process. Therefore, I maintain my rating (3, Reject).
> > > >
> > > >
> > > > [Brix et al., 2023]: Christopher Brix, Stanley Bak, Changliu Liu, Taylor T. Johnson: The Fourth International Verification of Neural Networks Competition (VNN-COMP 2023): Summary and Results. CoRR abs/2312.16760 (2023)
> > > >
> > > > [Müller et al., 2022] Mark Niklas Müller, Christopher Brix, Stanley Bak, Changliu Liu, Taylor T. Johnson: The Third International Verification of Neural Networks Competition (VNN-COMP 2022): Summary and Results. CoRR abs/2212.10376 (2022)
> > > >
> > > > [Zhang et al., 2022]: Huan Zhang, Shiqi Wang, Kaidi Xu, Linyi Li, Bo Li, Suman Jana, Cho-Jui Hsieh, J. Zico Kolter: General Cutting Planes for Bound-Propagation-Based Neural Network Verification. NeurIPS 2022.

---

> > > > > ### Author Response · Authors · 2024-08-14
> > > > >
> > > > > Dear reviewer aDbJ,
> > > > >
> > > > > Thank you for engaging in this discussion. We are also adding a comparison with GCP-CROWN [2]. We believe that the other concerns raised in the final comment have already been addressed in our previous responses. However, we would like to provide a summary of those responses once more.
> > > > >
> > > > > **Q1: Non-relational baselines.**
> > > > >
> > > > > **R1:** As instructed, we have already provided a comparison with MNBaB [1], demonstrating that our proposed method, RABBit, significantly outperforms it. We are also adding a comparison with GCP-CROWN [2]. We compared the performance of RABBit with GCP-CROWN across all ConvSmall CIFAR10 networks listed in Table 1 of the paper. The comparison used the same $\epsilon$ values, hardware, and timeout values as mentioned in Section 6.1. While GCP-CROWN outperforms $\alpha,\beta$-CROWN on all networks, it remains significantly less precise than RABBit, as shown in Table 10.
> > > > >
> > > > > Table 10 UAP accuracy of RABBit vs baselines including MNBaB and **GCP-CROWN**
> > > > > | Training         |   $\epsilon$ | $\alpha$-CROWN| RACoon |   $\alpha,\beta$-CROWN |   MNBaB |GCP-CROWN |  RABBit |
> > > > > |:-----------------|-------:|:-------------|------:|------:|------:|---------:|---------:|
> > > > > | DiffAI         |  5/255  | 49.6%|  51.6%|  53.6%|  55.0% |53.8%  | **59.8%**   |
> > > > > | SABR   | 2/255   | 75.8% | 78.2% | 78.4% | 80.0% | 80.0% |  **84.0%**  |
> > > > > | CITRUS | 2/255   | 76.0%| 78.8% | 79.0%  | 79.6% | 79.6%  |  **83.6%**  |
> > > > > | Standard      | 1/255    | 45.4% | 45.4%  | 59.8%  | 55.0%  |61.8% | **62.4%**  |
> > > > >
> > > > >
> > > > >
> > > > >
> > > > > **Theoretical limitation of non-relational verifiers:**
> > > > > Beyond experimental comparisons, any non-relational verifier, including GCP-CROWN [2], developed for verifying input-specific attacks, is limited in verifying relational properties because it ignores cross-executional input dependencies (see lines 168–174 in the paper).
> > > > >
> > > > > **Q2: Sensitivity of the results**
> > > > >
> > > > > **R2:** It is expected that the verified worst-case UAP accuracy for different sets of images of size $k=50$ can be different. However, as shown in Tables 6 and 7 of the rebuttal, for ConvSmall CITRUS [3] for CIFAR10 and MNIST respectively, RABBit significantly outperforms the baselines across all quantiles. Additionally, we compute the difference between RABBit's and $\alpha,\beta$-CROWN's UAP accuracy for ConvSmall CITRUS [3] for both CIFAR10 and MNIST and report the mean, 95% confidence interval, and median in Table 11. The 95% confidence interval substantiates RABBit obtains **statistically significant** improvement over the baseline.
> > > > >
> > > > > Table 11: Difference (%) between RABBit and $\alpha,\beta$-CROWN's UAP accuracy for ConvSmall CITRUS[3]
> > > > >
> > > > > | Dataset         |   $\epsilon$ |  Mean |   95% CI |   Median |
> > > > > |:-----------------|-------:|:-------------|------:|------:|
> > > > > |CIFAR10| 2.0| 4.6| [1.8, 7.4] | 5.0|
> > > > > |MNIST| 0.15| 2.2| [0.9, 3.5] | 2.0|
> > > > >
> > > > > In our evaluation, we run RABBit and baselines on 10 UAP properties each with k = 50 inputs, totaling **500 inputs**. Many DNN verifier papers, such as Refinezono [4], deepG [5], RACoon [6],  and GCP-CROWN [2], **only use 100 inputs** for evaluation. Moreover, the SOTA relational DNN verifier RACoon [6] **also considers 10 UAP properties** in the evaluation section of its paper.
> > > > >
> > > > > [1] "Complete Verification via Multi-Neuron Relaxation Guided Branch-and-Bound", C. Ferrari, et. al., ICLR, 2022.
> > > > >
> > > > > [2] "General Cutting Planes for Bound-Propagation-Based
> > > > > Neural Network Verification", Zhang, et. al., NeurIPS, 2022.
> > > > >
> > > > > [3] "Cross-Input Certified Training for Universal Perturbations", C. Xu, et. al., ECCV, 2024.
> > > > >
> > > > > [4] "Boosting Robustness Certification of Neural Networks", Singh, et. al., ICLR, 2019.
> > > > >
> > > > > [5] "Certifying Geometric Robustness of Neural Networks", Balunovic, et. al., NeurIPS, 2019.
> > > > >
> > > > > [6] "Relational DNN Verification With Cross Executional Bound Refinement", Banejee, et. al., ICML, 2024.

---

> > > > > > ### Author Response · Authors · 2024-08-14
> > > > > >
> > > > > > **Q3: As was discussed, Algorithm 1 does not properly reflect the actual implementation of RABBit. This leads to inconsistencies, for example, with regard to the presented timeout mechanism and the actually deployed timeout mechanism.**
> > > > > >
> > > > > > **R3:** As detailed in the paper, the timeout for RABBit and all baselines is set to $k$ minutes, consistent with Algorithm 1. We have already provided the GPU specifications used for BaB and the CPU specifications used for MILP in the paper. Running each CPU-bound MILP optimization in a separate thread allows us to avoid unnecessarily blocking the GPU-bound BaB computation, while still utilizing the same CPU resources available to all baselines. Additionally, as discussed above, all these intermediate MILPs can be replaced with a single MILP that incorporates all available constraints before $(T_{total} - T_{MILP})$. This alternative approach, as demonstrated in Table 9 of the rebuttal, yields results comparable to those in Table 1 of the paper. We will clarify these details in the next revision.
> > > > > >
> > > > > >
> > > > > >
> > > > > >
> > > > > > **Q4: Reproducibility of the code .**
> > > > > >
> > > > > >
> > > > > > **R4:** In the README of the code, we provide detailed instructions on how to install dependencies, all the pre-trained models used for experiments, detailed descriptions of the different hyperparameters used by RABBit and the baselines, various test cases to reproduce the main results and ablations, and instructions on how to run new experiments.
> > > > > >
> > > > > > For instance, the following two commands will reproduce the results for the CIFAR10 ConvSmall DiffAI network from Table 1.
> > > > > >
> > > > > > 1. `python -m unittest -v tests.test_rabbit.TestUAP.test_cifar_uap_diffai`
> > > > > >
> > > > > > 2. `python -m unittest -v tests.test_results.TestResults.test_analyze_main_results_cifar_diffai`
> > > > > >
> > > > > >
> > > > > > These two commands needed to run the experiment and analyze the result are already described in the README of the provided code.

---

### Official Review · Reviewer_dFSH · 2024-07-08

**Soundness:** 3
**Presentation:** 1
**Contribution:** 3
**Rating:** 6
**Confidence:** 4

**Summary:**

The paper presents RABBIt, a general framework for improving the precision of relational verification of DNNs through BaB methods and cross-execution dependencies. RABBIt uses strong bounding and strong branching methods, outperforming BaB methods like $\alpha$-$\beta$-CROWN and RACoon, which only leverages cross-execution dependencies. Experiments show that RABBIt significantly outperforms SOTA verifiers for relational properties on various architectures trained by different training methods.

**Strengths:**

1. Strong bounding and strong branching methods are technically solid.
2. Experiments show that RABBIt significantly outperforms SOTA verifiers for relational properties on various architectures trained by different training methods. The paper also presents extensive ablation studies on strong branching, strong bounding, and other hyper-parameters.

**Weaknesses:**

1. The paper claims that strong bounding provides tighter bounds than RACoon and $\alpha$-$\beta$-CROWN. Although the experiments validate this claim, the paper offers no proof. This claim might not hold, especially considering the $k_t$ and the time limit used in Algorithm 1.
2. Another concern is the poor presentation of this paper.
   * In Lines 49-50, the bullet points a), b), and c) do not correspond to the order of challenges introduced earlier in the paragraph. c) should come first, then b), and finally, a).
   * In Line 179, the MILP instance appears abruptly. It is confusing because the reader might think of it as the MILP problem introduced in Line 36. The purpose of the MILP instance only becomes clearer after introducing the k-UAP accuracy problem.
   * However, the k-UAP accuracy is not adequately introduced and abruptly appears in Line 227.
   * In lines 221-224, I cannot follow how to derive $m^{1/n}$ and $m/n$. Please give me an example.
   * In Section 4.2, $N(x_i+\delta)$ is a vector but $L_j^T(x_i+\delta)$ is a scalar. Does the paper want to express $c_i^T N(x_i+\delta)$?
   * In Section 4.2, the strong branching method seems to me only a preprocessing rather than a counterpart to the strong bounding method. Is the strong branching method only to guarantee that RABBIt's bound is tighter than $\alpha$-$\beta$-CROWN bound?
   * The presentation in Section 4.2 could be better. For example, the paper should relentlessly remind readers that the goal of Lines 243-267 is to compute the lower bound $b_j^*$.
   * In Line 319, the index $j$ is not quantified.
   * In Line 336, Wang et al. should be non-relational, and Banerjee and Singh should be relational.

**Questions:**

1. In lines 221-224, I cannot follow how to derive $m^{1/n}$ and $m/n$. Please give me an example.
2. In the MILP encoding of RABBIt, What are the new terms/constraints compared to RACoon?
3. Is the strong branching method only to guarantee that RABBIt's bound is tighter than $\alpha$-$\beta$-CROWN bound?

Comments:
1. Missing related work, [1] provides verification for counterfactual explanation robustness, which is also a relational property of neural networks.

[1] Verified Training for Counterfactual Explanation Robustness under Data Shift. Anna P. Meyer, Yuhao Zhang, Aws Albarghouthi, Loris D'Antoni

**Limitations:**

The paper discusses its limitations.

---

> ### Author Rebuttal · Authors · 2024-08-07
>
> **Q1: In lines 221-224, I cannot follow how to derive $m^{1/n}$ and $m/n$. Please give me an example.**
>
>  **R1:** Please refer to the answer to Q3 in the common response.
>
> **Q2: In the MILP encoding of RABBIt, What are the new terms/constraints compared to RACoon?**
>
> **R2:** The MILP encoding in RABBit includes additional constraints derived from both strong bounding and strong branching, which are not utilized in RACoon.
> - **Strong branching constraints:** For each target $L_{i}^{j}$, we add a new lower bound constraint $o_i \geq L_{i}^{jT}(\pmb{x_i} + \pmb{\delta}) + b^{j}_{i}$ where $b^{j}\_{i}$ is obtained by strong branching and $o_i = \pmb{c_i}^{T}N(\pmb{x_i} + \pmb{\delta})$. This is discussed in the line 319 of the paper.
> - **Strong bounding constraints:** Suppose for any subset of executions $S \subseteq [k]$, strong bounding proves the absence of common perturbation. We add the following lower bound constraint $\sum_{i \in S} z_{i} \geq 1$. Here $z_i = (o_i \geq 0)$ is a binary variable that indicates whether the constraints $\pmb{c_i}^{T}N(\pmb{x_i} + \pmb{\delta}) \geq 0$ is satisfied or not. The lower bound constraint $\sum_{i \in S} z_{i} \geq 1$ follows from the fact for any $\|\pmb{\delta}\|_{\infty} \leq \epsilon$ at least one execution from $S$ remains correctly classified. This is discussed in the lines 313 - 316 of the paper.
>
>
> **Q3: Strong bounding provides tighter bounds than RACoon and $\alpha,\beta$-CROWN. Although the experiments validate this claim, the paper offers no proof. This claim might not hold, especially considering the $k_t$ and the time limit used in Algorithm 1.**
>
> **R3:** Strong bounding is a BaB method that uses cross-executional bounding as the bounding step. Given RACoon uses the same bounding methods and does not employ branching, for the same time limit, strong bounding will be at least as precise as RACoon. However, as correctly pointed out, for some cases $\alpha,\beta$-CROWN can outperform strong bounding. However, our experiments show that for practical scenarios, strong bounding outperforms $\alpha,\beta$-CROWN.
>
> **Q4: In Section 4.2, the strong branching method seems to me only a preprocessing rather than a counterpart to the strong bounding method. Is the strong branching method only to guarantee that RABBIt's bound is tighter than $\alpha,\beta$-CROWN bound?**
>
> **R4:** Strong bounding can only prove the absence of common perturbation for a set of executions $S \subseteq [k]$. Hence, with strong bounding, we can only show at least 1 execution from $S$ is correctly classified. However, this approach is suboptimal for cases where the number of correctly classified executions from $S$ is $> 1$. In contrast, strong branching extracts the same target linear approximation for all subproblems, allowing us to formulate an efficiently optimizable MILP instance with these targets. This MILP instance can address cases where more than one execution from $S$ is correctly classified.
> Also, strong branching allows us to explore more branches per execution as explained in answer to Q3 in the common response. Strong branching is not a preprocessing step and is important for improving the precision of RABBit, as demonstrated by the results in Table 1 of the paper.
>
> **Q5: Missing related work.**
>
> **R5:** Thanks for pointing out. We will add it to the revised version of the paper.
>
> **Q6: In Section 4.2, $N(\pmb{x_i} + \pmb{\delta})$ is a vector but $\pmb{L_{j}}^{T}(\pmb{x_i} + \pmb{\delta}) + b^{*}\_j$ is a scalar. Does the paper want to express $\pmb{c_{i}^{T}}N(\pmb{x_i} + \pmb{\delta})$?**
>
> **R6:** Thanks for pointing out. In section 4.2, it should be $\pmb{c_{i}^{T}}N(\pmb{x_i} + \pmb{\delta})$ instead of $N(\pmb{x_i} + \pmb{\delta})$. We will correct it in the revised version of the paper.
>
> **Q7: Presentation of the paper.**
>
> **R7:** We will provide a formal definition of worst-case UAP accuracy and the MILP optimization problem used to compute it before they are cited. We will also correct the citation issues noted in the revised version.

---

> > ### Comment · Reviewer_dFSH · 2024-08-07
> >
> > Thank you for the response. It addresses all my concerns.

---

> > > ### Author Response · Authors · 2024-08-14
> > >
> > > Thanks for your feedback. We will add your suggestions to strengthen the paper in the revised version.

---

### Official Review · Reviewer_uEoH · 2024-07-10

**Soundness:** 3
**Presentation:** 3
**Contribution:** 3
**Rating:** 7
**Confidence:** 4

**Summary:**

The paper proposes a branch and bound technique for the relational verification of neural networks.
To this end, the authors build upon RACoon (Banerjee and Singh [2024]) and $\alpha,\beta$-CROWN (Wang et al. [2021b]).
The work describes two branching mechanisms (strong bounding and strong branching) that form a symbiosis
in the authors' new algorithmic framework RABBit.

Referenced citations below:
- [R1] Sälzer, Marco, and Martin Lange. "Reachability in Simple Neural Networks." Fundamenta Informaticae 189.3-4 (2022): 241-259.

**Strengths:**

(S10) The authors address an important problem since universal adversarial perturbations are a much stronger attack-vector than instance-specific adversarial attacks.

(S20) While the idea initially seems very straight-forward, the paper introduces interesting and subtle concepts with its distinction between strong branching and bounding which are
      shown to be useful in practice. In particular, the paper's approach is, to the best of my knowledge, novel.

**Post Rebuttal:**
I consider this distinction between strong branching and bounding, together with the experiments that show the approach scales to a large number of compared executions, the major contribution of the paper.

(S40) The paper is mostly well-written and provides a good overview on the approach.

**Weaknesses:**

Major Opportunity for Improvement:

(O10) In the introduction of the evaluation the paper states "We evaluate [..] on multiple relational properties, DNNs, and datasets."
      Unfortunately, the paper's evaluation focuses entirely on the k-UAP property. While this is an interesting property worth studying,
      it would be necessary to evaluate the approach on additional relational properties to substantiate the claim that the tool is a generally applicable relational verification tool.
      At least, it would be helpful to discuss necessary modifications to adapt the approach to other relational properties.
      Alternatively, the paper's title and story could be focused on Universal Adversarial Perturbation.

**POST REBUTTAL:**
The authors provided experiments on a second relational property.
The results seem less impressive to me in this case. Nonetheless, this demonstrates the applicability to other relational properties.
I get the impression the approach is particularly valuable when comparing a large number of executions (as done for k-UAP with k>=20 in the paper).

In my initial comment yesterday I said I would raise my score to Weak Accept if the code submission problems are resolved.
I gave this some more thought:
While the results are less impressive in the new experiments, I do believe the community is better off with a verifier that can solve relational properties for large $k$ than it is without such a verifier. Hence, I have decided to raise my score to **Accept**.

Minor Opportunities for Improvement:

(O1) In Line 223 you mention that simultaneously branching over all executions leads to the processing of only $m^{\frac{1}{n}}$ subproblems.
     It would be helpful if you provide some more intuition to the reader about this. My understanding is that we obtain this number, because
     a split for one execution effectively splits the state space for all executions and thus if we split each execution into l subproblems this
     leads to l^n (for n executions) subproblems that have to be handled. This could be said more explicitly here.

**POST REBUTTAL:**
The authors proposed sufficient improvements.

(O2) In Line 324 you note that "all complete non-relational verifiers are also incomplete for relational properties since they do not track any dependencies between executions."
     In this generality, this statement seems factually wrong. Obviously, completeness also depends on the checked property, however there is
     a large class of properties which can be encoded as linear constraints over a product network. In this case, I see no reason why the completeness guarantee of a non-relational
     verifier should not equally hold for (linear constraint based) relational properties.
     For example Appendix A.1 describes how k-UAP can be formalized via linear constraints.
     While dependency tracking is very useful for efficient relational NN verification, it does not seem to be strictly necessary.
     Can the heuristics employed by RaBBit and RACoon lead to incompleteness when it comes to providing exact UAP bounds?

**POST REBUTTAL:**
I am not convinced of the authors' answer in this respect.
In Appendix A.1 the authors describe how the k-UAP property can be encoded with linear constraints. Maybe I am missing something here, but even with the authors' clarification, it is not clear to me why verifying this specification on a product NN would not yield a complete verifier for k-UAP. Again, I wholeheartedly agree that tracking relational dependencies is *useful*, but I still do not see why it is *necessary*. As stated in the authors' response (Q3), completeness for UAP is then "just" a matter of checking all possible subsets (which is again an issue independent of dependency tracking).

**POST COMMENTS:**
In the comments the authors have sufficiently resolved this issue: Indeed, the constraints outlined in A.1 already are cross-executional constraints.

Minor notes:
- Line 52/53: "with *a* cross-executional bound method"
- Line 104: DNN verification is not only NP-hard, but actually (exactly) NP-complete [R1]
- Line 157: "...it is possible to *statistically* estimate..."
- Line 336: You mixed up the citations for relational / non-relational verifiers here

**Questions:**

(Q1) Do I correctly assume that your comparison with alpha-beta-CROWN uses a problem formulation based on a product NN and the k-UAP linear constraint formulation?

**POST REBUTTAL:**
The authors clarified this.

(Q2) Do you have experiments w.r.t. other relational properties? If not, would you be willing to adjust the title and content of the paper to focus on UAP properties? -- In either case, I would be willing to raise my score to Accept.

**POST REBUTTAL:**
See (O10)

**Limitations:**

Concerning limitations, my main critique is the focus on Universal Adversarial Perturbations in the experimental section (see (O10)).

**POST REBUTTAL:**
Due to a lack of access to NVIDIA GPUs, I was not able to evaluate the code submission.
However, aDbJ has confirmed that the (updated) code submission allows execution of the code.

---

> ### Author Rebuttal · Authors · 2024-08-07
>
> **Q1: Do you have experiments w.r.t. other relational properties? If not, would you be willing to adjust the title and content of the paper to focus on UAP properties? -- In either case, I would be willing to raise my score to Accept.**
>
> **R1:** Please refer to the answer to Q1 in the common response.
>
> **Q2: "All complete non-relational verifiers are also incomplete for relational properties since they do not track any dependencies between executions" - Completeness depends on the specific relational property.**
>
> **R2:** We agree that completeness depends on the specific relational property being verified. In this case, we refer to relational properties such as robustness against UAP, where it is important to track dependencies between perturbed inputs across different executions to achieve completeness. Existing complete verifiers like $\alpha,\beta$-CROWN, which are designed for verifying input-specific robustness, will be incomplete if they verify each execution independently without tracking these dependencies. Please refer to lines 168-173 of the paper, where we describe a scenario in which input-specific adversarial perturbations exist for individual inputs, but no common adversarial perturbation exists. We will rephrase the highlighted sentence to clarify this.
>
> **Q3: Can the heuristics employed by RABBit and RACoon lead to incompleteness when it comes to providing exact UAP bounds?**
>
> **R3:** RACoon does not use any branching and is incomplete. In RABBit, for scalability, we greedily select subsets of executions and run strong bounding only on the selected subsets. For this reason, the current implementation of RABBit is also incomplete and only computes a sound lower bound on the worst-case UAP accuracy. However, it is possible to make RABBit complete by considering all possible subsets of executions and then formulating the MILP formulation.
>
> **Q4: Do I correctly assume that your comparison with $\alpha,\beta$-CROWN uses a problem formulation based on a product NN and the k-UAP linear constraint formulation?**
>
> **R4:** Yes, $\alpha,\beta$-CROWN is applied to the product DNN. However, unlike the strong bounding approach proposed in RABBit, the bounding method of $\alpha,\beta$-CROWN does not track any dependencies. As a result, even when $\alpha,\beta$-CROWN is executed on the product DNN, it computes the lower bound $\pmb{c_i}^TN(\pmb{x_i} + \pmb{\delta})$ for each execution independently and loses precision.
>
> **Q5: Questions about $m^{1/n}$ and $m/n$ subproblems.**
>
>  **R5:** Please refer to the answer to Q3 in the common response. We will add a more explicit description in the revised version.
>
> **Q6: Minor notes on the presentation of the paper.**
>
> **R6:** Thanks for pointing out. We will add the suggested changes in the revised version of the paper.

---

> > ### Comment · Reviewer_uEoH · 2024-08-08
> > **Questions mostly addressed -- Code problem needs to be addressed**
> >
> > Dear authors,
> >
> > thank you for adressing my questions.
> >
> > ### Concerning (O10)/(Q1):
> > I appreciate the additional experiments on a second relational property.
> > While the improvements over the SoTA look less impressive to me in this case, it is nice to see the approach is applicable to other properties.
> > I get the impression the approach is particularly valuable when comparing a large number of executions (as done for k-UAP with k>=20 in the paper).
> >
> > ### Concerning (O2)/(Q2):
> > In Appendix A.1 you describe how the k-UAP property can be encoded with linear constraints.
> > Maybe I am missing something here,
> > but even with your clarification it is not clear to me why verifying this specification on a product NN would not yield a complete verifier for k-UAP.
> > Again, I wholehearthedly agree that tracking relational dependencies is *useful*, but I still do not see why it is *necessary*.
> > As stated in your response (Q3) completeness for UAP is then "just" a matter of checking all possible subsets (which is again an issue independent of dependency tracking).
> >
> > ### Concerning (O1)/(Q5):
> > Thanks for the clarification which addresses my concern.
> >
> > ### Code
> > I promised to raise my score if (O10)/(Q1) is addressed and I am still willing to do increase to Weak Accept.
> > However, I am now somewhat worried about the state of the code submission given the remarks by aDbJ.
> > Following up on aDbJ's review, I have also tried to execute the original code submission:
> > While I was able to fix the mentioned issues, the test can indeed not be performed without the NNs available.
> > To make matters worse (and I do not consider this a fault of the authors), it seems that Google Drive has removed the updated code submission.
> > Maybe, an anonymous figshare could help here. I sincerely hope this problem can be solved.
> >
> >
> > ### Minor notes:
> > line 157: "...it is possible to *statistically* estimate..."

---

> > > ### Author Response · Authors · 2024-08-09
> > >
> > > Dear reviewer uEoH,
> > >
> > > Thanks for your response. We have added clarifications to your questions and will be happy to help if you face any issues with the code.
> > >
> > > **Q1: The proposed approach has higher gains when comparing a large number of executions .**
> > >
> > > **R1:** We evaluate the performance of RABBit with different values of $k$ in Figure 4 and Figure 7 of the paper. RABBit's performance improvement grows with higher $k$ values, which is anticipated. At a high level, as the size of the set of executions increases, it becomes more challenging to find common adversarial perturbations. This helps RABBit which exploits cross-executional dependencies.
> > >
> > > **Q2: Even with your clarification it is not clear to me why verifying this specification on a product NN would not yield a complete verifier for k-UAP ?**
> > >
> > > **R2:** We believe there is a misunderstanding regarding what we meant by tracking dependencies between perturbed inputs. The input specification $\Phi = \bigwedge_{i=1}^{k} \phi_{in}^{i} \bigwedge \Phi^{\delta}$ (line 563 of the paper) for the $k$-UAP property includes the cross-executional input constraint $\Phi^{\delta}$, which defines the relationship between perturbed inputs from different executions. A verifier that ignores the cross-executional input constraints $\Phi^{\delta}$ would still be sound for $k$-UAP by verifying against the weakened input specification $\Phi' = \bigwedge_{i=1}^{k} \phi_{in}^{i}$. However, such verifiers would not be complete. In contrast, as correctly pointed out, verifiers that utilize the full input specification $\Phi$ (as opposed to $\Phi'$) on a product NN and use the constraint $\Phi^{\delta}$ can achieve completeness for $k$-UAP (as illustrated in our response to Q3). We will clarify this in detail in the revised version of the paper.
> > >
> > > **Q3: Link the code and networks used for experiments.**
> > >
> > > **R3:** We apologize for the missing networks and the faulty link. We have updated the link to our code in the previous thread and are also providing the link here. The networks used for the experiments can be found in the `RABBit/RABBit-main/nets` folder. We have added instructions for reproducing the experiments in the README file. If you encounter any issues while reproducing the code, we will be happy to assist you.
> > >
> > > anonymized link to the code: https://figshare.com/s/9dfe74654ea6f5a5ee24

---

> > > > ### Author Response · Authors · 2024-08-14
> > > >
> > > > Thank you for your feedback and for raising your score. We will incorporate your suggestions in the revised version of the paper.

---

### Official Review · Reviewer_eacX · 2024-07-13

**Soundness:** 3
**Presentation:** 1
**Contribution:** 3
**Rating:** 7
**Confidence:** 5

**Summary:**

This paper addresses the problem of verifying certain DNN properties that depend on multiple executions of the DNN, known as “relational verification.” The example used throughout the paper is verifying the k-UAP problem, which aims to confirm the absence of a universal adversarial perturbation for a given DNN. This problem requires relational verification because in order to compare the perturbations across multiple inputs, there must be multiple executions of the DNN considered during the verification process. This introduces additional difficulties compared to traditional single execution branch and bound techniques. The authors tackle this problem by introducing a “product DNN”, which is essentially duplicate DNNs for each of the executions in consideration, and propose stronger branching and bounding algorithms on this DNN instance. The results from the branching and bounding algorithms are then used in an MILP formulation, which they call RABBit. The experimental evaluation across pre-trained DNNs on MNIST and CIFAR-10 demonstrate that RABBit outperforms traditional DNN verifiers as well as the individual branching or bounding algorithms independently, highlighting the importance of combining the strengths of both.

**Strengths:**

The paper addresses an interesting problem. The comparison with prior approaches demonstrates that the proposed approach works well.

**Weaknesses:**

The paper suffers from a number of grammatical errors, particularly run-on sentences (e.g., page 4 line 171)  and improper use of commas which makes the paper difficult to read. Additionally, the juxtaposition of the actual contributions of the paper compared to prior work in verification is not always clear.

Another major issue throughout the paper is the presentation of “relational verification” as a concept, while only one example of the use case of relational verification (UAP verification) is presented. For example, on page 1 line 21, “relational properties common in practical situations” is never expanded on—what are the practical situations besides UAP? On page 8 line 356, “10 relational properties” are mentioned but are not defined in the paper. Are there 10 different versions of UAP or are there 10 unique use cases of relational verification in general?

## Minor notes

- Page 2 line 80: “constructing ‘product DNN’” → “constructing a ‘product DNN’”
- Page 3 line 123: “upper bound that contain…” → “upper bounds that contain”
- Page 4 line 184: “Let for all i \in S…” → “For all i \in S, let…”
- Page 4 line 186: “proves absence of common perturbation” → “proves the absence of a common perturbation”
- Page 4 line 192: “Formally, product DNN is a function” → “Formally, a product DNN is a function”
- Page 5 line 216: “do not have common perturbation” → “do not have a common perturbation”
- Page 5 line 217: “The detailed proof in the Appendix B” → “The detailed proof is in Appendix B”
- Page 5 line 224: “m/n subproblem per execution” → “m/n subproblems per execution”
- Page 6 line 270: “Alog. 1” → “Algo. 1”

**Questions:**

- In Table 1, how long does it take to run each verification?
- In Table 1, why the non-relational verifier alpha-beta-crown outperformed the STOA relational verifier RACoon? Is it because alpha-beta-crown uses BaB?
- Are there relational properties other than UAP that can be verified and evaluated?

**Limitations:**

Yes, the authors addressed the limitations of their work.

---

> ### Author Rebuttal · Authors · 2024-08-07
>
> **Q1: In Table 1, how long does it take to run each verification?**
>
> **R1:** Please refer to the answer to Q2 in the common response.
>
> **Q2: In Table 1, why the non-relational verifier $\alpha,\beta$-CROWN outperformed the SOTA relational verifier RACoon? Is it because $\alpha,\beta$-CROWN uses BaB?**
>
> **R2:** SOTA relational verifier RACoon is incomplete and utilizes a single bounding step without branching. Although RACoon is significantly more precise than $\alpha$-CROWN, the bounding step used in $\alpha,\beta$-CROWN, $\alpha,\beta$-CROWN outperforms it by exploring a large number of branches with BaB. We discuss this limitation of RACoon in lines 47--48 of the paper.
>
>
> **Q3: Are there relational properties other than UAP that can be verified and evaluated?**
>
> **R3:** Please refer to the answer to Q1 in the common response.
>
> **Q4: Are there 10 different versions of UAP or 10 unique use cases of relational verification in general?**
>
> **R4:** We consider 10 different instances of the UAP verification problem where each instance is defined on a set of 50 images as described in line 356. For the details on another relational property that can be verified by RABBit, please refer to the response to Q1 in the common response.
>
> **Q5: Grammatical errors and typing mistakes.**
>
> **R5:** We apologize for the grammatical and typing errors and will correct them in the revised version of the paper.

---

> > ### Comment · Reviewer_eacX · 2024-08-12
> >
> > Thank you for the response that clarified my concerns.

---

> > > ### Author Response · Authors · 2024-08-14
> > >
> > > Thank you for your feedback and for raising your score. We will incorporate your suggestions in the revised version of the paper.

---

### Author Rebuttal · Authors · 2024-08-07

Dear Area Chair and Reviewers,

We appreciate the constructive feedback from the reviewers and are encouraged by their acknowledgment of the paper's  theoretically sound contributions, and detailed experimental validation.

 ***Q1: Evaluating RABBit on other relational properties (eacX, uEoH)***

**R1:** We evaluate RABBit w.r.t another relational property: top-$k$ accuracy.

**Definition:** Given an unperturbed input $\pmb{x}$, a network $N: \mathbb{R}^{n_0} \rightarrow \mathbb{R}^{n_l}$, and $k \leq n_l$, we want to verify whether the target class remains within the top-$k$ classes predicted by $N(\pmb{x} + \pmb{\delta})$ for all bounded perturbations $\|\pmb{\delta}\|_{\infty} \leq \epsilon$.

Let $d_i(\pmb{\delta}) = \pmb{c_i}^TN(\pmb{x} + \pmb{\delta}) = N(\pmb{x} + \pmb{\delta})[t] - N(\pmb{x} + \pmb{\delta})[i]$ denote the logit difference between the target class $t$ and another class $i \neq t$. If, for all $|\pmb{\delta}|_{\infty} \leq \epsilon$, at least $n_l - k$ logit differences are positive, then we prove that the target class always remains within the top-$k$ predicted classes. For $i, j \in [n_l]$ and $i \neq j$, since the logit differences $d_i(\pmb{\delta})$ and $d_j(\pmb{\delta})$ are related, tracking their dependencies improves precision. In this case, even though all logit differences result from perturbations of the same input, we can treat the computation of each logit difference as a separate execution of $N(\pmb{x} + \pmb{\delta})$. This approach reduces the top-$k$ verification problem to a relational verification problem, which is handled by RABBit. Existing non-relational verifiers [1] compute the lower bound on each logit difference independently and thus lose precision.

In Table 1, we present results for top-$2$ ($k=2$) accuracy for ConvSmall DiffAI networks and $\epsilon$ values from Table 1 of the paper. We use the first 100 images from each dataset. The timeout values used are 1 minute for BaB per image and 1 minute per MILP instance. Since all the related logit differences originate from the same image, we do not need to greedily select executions for strong bounding. In all cases, RABBit is more precise than all the baselines. We will include results for all the networks in the revised version of the paper.

Table 1: Verified top-2 accuracy for RABBit vs baselines
|Dataset | Training         |   $\epsilon$ | $\alpha$-CROWN| Avg. Time (s.) | RACoon| Avg. Time (s.) |   $\alpha,\beta$-CROWN |   Avg. Time (s.) |  RABBit | Avg. Time (s.) |
|:-----------------|-------:|-------:|:-------------|------:|------:|------:|---------:|---------:|---------:|---------:|
|CIFAR10| DiffAI | 5/255 |74% | 4.52|75% | 4.87|75%|20.47|**78%**|24.27|
|MNIST| DiffAI | 0.13 |84% | 1.20|84% |1.42|89%|11.03|**91%**|13.43|

[1] "Beta-CROWN: Efficient Bound Propagation with Per-neuron Split Constraints for Neural Network Robustness Verification", S. Wang, et. al., NeurIPS, 2021.

***Q2: Verification Times (eacX, aDbj)***

**R2:** We present the runtimes, in seconds, for each verifier averaged across all properties and networks from Table 1 of the paper in Tables 2 and 3 below. For each property, the runtime of a verifier is the timestamp when the verifier first achieves its maximum UAP accuracy within the time limit. Although RABBit has a higher runtime overhead compared to existing verifiers such as $\alpha,\beta$-CROWN, it consistently delivers better performance at every timestamp, as shown in Figure 1 of the paper. We will add these results to the revised version of the paper.

  Table 2: Runtime statistics of different verifiers for CIFAR-10
  | Verifier | Mean | 95% CI | Std. | Min| 25%| Median | 75%| Max|
|----------|----------|----------|----------|----------|----------|----------|----------|----------|
| α-CROWN    | 4.3   | [2.9, 5.7] | 5.4 | 0.3 | 0.9 | 1.6 | 6.1 | 27.1|
| α,β-CROWN   | 1325.6   | [1220.8, 1430.5] | 402.5 | 418.5 | 990 | 1363.6 |1608.2 | 2199.2|
| RACoon   | 409.1   | [353.1, 464.9] | 214.6 | 66.2 | 260.6 | 342.4 | 607.9 | 944.1 |
| RABBit  | 1725.5   | [1491.5, 1959.5] | 898.3 | 178 | 1064.3 | 1528.3 | 2451.1 | 2998 |

  Table 3: Runtime statistics of different verifiers for MNIST
  | Verifier | Mean | 95% CI |Std. | Min| 25%| Median | 75%| Max|
|----------|----------|----------|----------|----------|----------|----------|----------|----------|
| α-CROWN    | 2.9   | [1.6, 4.2] | 4.5 | 0.3 |0.5 | 0.8 | 4.4 | 20|
| α,β-CROWN   | 1625.5   | [1409.4, 1849.6] | 752.8 | 227.7 | 1075.7| 1389 | 2104 | 2943.8|
| RACoon   | 604.8   | [539.0, 670.6] | 229| 18.8| 389.5 | 658.8 | 807.6 | 981.6 |
| RABBit  | 2083.7 | [1857.2, 2310.2] | 788.9| 557.3 | 1697.5 | 2120.4 | 2783.2 | 2999.1 |

**Q3: Derivation of m/n and m^(1/n) (uEoH, dFSH)**

**R3:** Suppose we consider a relational property over 4 executions ($n = 4$) and in the given time thershold we can solve $16$ problems ($m = 16$).
Now for strong bounding, since we are tracking dependencies across executions we need to consider all possible combinations of subproblems from different executions. Assuming we split on each execution uniformly we can only consider $2$ subproblems ($m = 2^n$) from each execution. If $(A_1, A_2)$, $(B_1, B_2)$, $(C_1, C_2)$, and $(D_1, D_2)$ are subproblems from each of 4 executions respectively then strong bounding considers the 16 subproblems specified by $(A_i, B_j, C_k, D_l)$ where $i, j, k, l \in $ {1, 2}.

In contrast, if we apply BaB on each execution independently we **do not** need to consider combinations of subproblems from different executions. In this case, assuming we split on each execution uniformly we can consider $4$ subproblems ($m = 4*n$) from each execution.

---

> ### Author Response · Authors · 2024-08-14
>
> Dear AC and reviewers,
>
> As the discussion period comes to an end, we want to express our gratitude for your time. We believe that our comments have addressed all concerns or questions. We are summarizing the clarifications to the major concerns here, and we hope that they will be taken into account when making the final decision.
>
> - **Code** - We have fixed the installation issues and provided all networks and detailed instructions for reproducing the experimental results. Thanks, reviewer aDbJ for confirming that the updated version of code is working.
> - **non-relational verifiers:** $\alpha,\beta$-CROWN is one of the most popular and strongest non-relational verifiers, and our submitted paper already includes a detailed comparison with it.  However, as suggested by reviewer aDbJ, we included comparisons with **both** MNBaB and GCP-CROWN on CIFAR10 networks. In both cases, RABBit significantly outperforms the corresponding non-relational verifier (see Table 10 of the rebuttal). Beyond the experiments, we discuss why non-relational verifiers are theoretically limited for relational properties. We append these additional results and explanations to further substantiate our claims.
> - **Sensitivity and experimental setup:** We show that the gains obtained by RABBit are statistically significant (see Table 11 of the rebuttal). Furthermore, the papers on various DNN verifiers, including GCP-CROWN and RACoon, use the same or even fewer inputs and relational properties in their evaluations. We also clarified that the experimental setup, including timeout values, matches what was described in the paper. We have not changed the experimental setup and proposed algorithm.
> - **Comment on Resubmission by Reviewer aDbJ:** It is a standard practice at top-tier ML conferences like NeurIPS, ICML, and ICLR to provide extra results/details as requested by the other reviewers. For example, this was the case for both GCP-CROWN [1] and MnBab [2] which the reviewer asked us to compare against. Moreover, issues related to software installation are not uncommon, even for widely used tools like alpha-beta CROWN [3] and ERAN [4], which have publicly documented installation problems on GitHub. We have already resolved the installation issues in our code, and reviewer aDbJ confirmed that the updated version of the code works. Since the concerns have already been addressed, incorporating the changes does not necessitate resubmission, as with other papers.
>
> [1] https://openreview.net/forum?id=5haAJAcofjc
>
> [2] https://openreview.net/forum?id=l_amHf1oaK
>
> [3] https://github.com/Verified-Intelligence/alpha-beta-CROWN/issues
>
> [4] https://github.com/eth-sri/eran/issues

---

### Author Response · Authors · 2024-08-07
**anonymized link to the code**

Dear AC,

As requested by reviewer aDbJ, we are submitting the following anonymized link to the code for reproducing our experiments:

Anonymized link: https://drive.google.com/file/d/1jIO2b76f9VDkCeZxaAJvnZxobXgGRE18/view?usp=sharing

---

> ### Comment · Reviewer_aDbJ · 2024-08-07
> **Code Still Buggy**
>
> The provided code is still buggy. The README in the zip archive linked above describes creating a conda environment with Python 3.7 (`RABBit-main/src/abc/complete_verifier/environment.yml`), but the code does not compile under Python 3.7
> ```
> python -m unittest -v tests.test_rabbit.TestUAP.test_cifar_uap_diffai
> Traceback (most recent call last):
>   File "/home/vboxuser/miniconda3/envs/rabbit2/lib/python3.7/runpy.py", line 193, in _run_module_as_main
>     "__main__", mod_spec)
>   [...]
>     from src.metaScheduler import MetaScheduler
>   File "/home/vboxuser/RABBit (copy)/RABBit-main/src/metaScheduler.py", line 565
>     return all_tuples, all_refined_tuples, orig_scores, *self.process_selected_tuples(selected_tuples, selected_refined_tuples)
> ```

---

> ### Author Response · Authors · 2024-08-08
>
> Dear Reviewers aDbJ and uEoH,
>
> It appears the issue may be related to the Python version you are using. Our experiments were conducted using Python 3.10.10. We have provided an updated version along with instructions on how to install the dependencies (see below and in the README file). After installation, please verify that the installed Python version is 3.10.10. Additionally, ensure that you have the Gurobi license installed on your device; instructions for obtaining the license are included in the README file.
>
> annonymized link to the code: https://figshare.com/s/9dfe74654ea6f5a5ee24
>
> ```
> # Remove any environment named rabbit if it exists
> conda remove -n rabbit --all
> # create the new environment from the RABBit directory
> conda env create -f environment_rabbit.yml --name rabbit
> # activate the environment
> conda activate rabbit
> # Check the correct python version "3.10.10" is installed or not
> python --version
>
> ```
>
> If you encounter any issues with the code, please let us know. We will be happy to assist you.

---

> > ### Comment · Reviewer_aDbJ · 2024-08-08
> >
> > I confirm that the new environment definition allows for executing the code.

---

### Decision · Program_Chairs · 2024-09-25

**Decision:**

Accept (poster)

**Comment:**

The reviewers appreciared the importance of the problem considered, and praised the novelty of the approach and the experimental results.  On the other hand, they struggled with the quality of the writing, and with running the code provided.  A number of issues identified by the reviewers were resolved by the rebuttal and the subsequent discussion with the authors.  On balance, I recommend acceptance, however urge the authors to revise the paper carefully based on all the reviews, and in particular to address apparent plagiarism or self-plagiarism in that a number of paragraphs in the paper are identical or almost the same as paragraphs in Banerjee and Singh, "Relational DNN Verification With Cross Executional Bound Refinement", ICML 2024.